# Retro-Expert: Collaborative Reasoning for Interpretable Retrosynthesis

Xinyi Li [* 1]   Sai Wang [* 1]   Yutian Lin [1]   Yu Wu [† 2]

## Abstract

Retrosynthesis prediction aims to infer the reactant molecules based on a given product molecule, which is a fundamental task in chemical synthesis. However, existing methods rely on a static pattern-matching paradigm, which limits their ability to perform effective logical decision-making from chemical data, leading to a black-box process. We propose **Retro-Expert**, an interpretable retrosynthesis framework that performs collaborative reasoning by combining the complementary strengths of Large Language Models and specialized models via pure reinforcement learning. It outputs natural language explanations grounded in chemical logic through three components: (1) specialized models provide chemical knowledge that is distilled into a high-quality chemical decision space, (2) LLM-driven critical reasoning to generate predictions with an interpretable reasoning path, and (3) knowledge-grounded policy optimization refines the interpretable decision policy. Experiments show that Retro-Expert surpasses both LLM-based and specialized models across different metrics, while generating chemically grounded explanations that enhance chemists' trust in practice. The source code for this paper is available at https://github.com/MagixRab-ll/Retro-Expert

## 1. Introduction

Retrosynthesis prediction aims to deduce potential reactants and reaction pathways for synthesizing a target product molecule (Segler & Waller, 2017; Somnath et al., 2021; Sun et al., 2021), holding significant value in drug discovery and molecular design (Wang et al., 2018; 2023; Hu et al., 2025). However, existing methods primarily adopt a pattern-matching paradigm, where chemical data is organized to

---
[*]Equal contribution [1]School of Computer Science, Wuhan University, China [2]School of Artificial Intelligence, Wuhan University, China. Correspondence to: Yu Wu <wuyucs@whu.edu.cn>.

*Proceedings of the 43rd International Conference on Machine Learning*, Seoul, South Korea. PMLR 306, 2026. Copyright 2026 by the author(s).

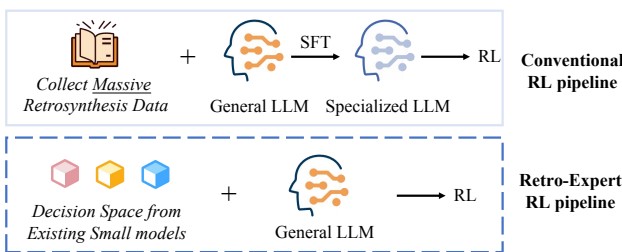

*Figure 1.* Comparison between the conventional RL pipeline and our method. Conventional methods require SFT as a necessary step for effective RL, whereas our method directly optimizes the LLM by leveraging the decision space.

learn mappings from product SMILES to reactant SMILES, framing the task as either classification or auto-regressive sequence generation (Zheng et al., 2019; Chen & Jung, 2021; Yao et al., 2024b). Under this paradigm, specialized model-based methods capture intricate structural features and exploit additional chemical priors to learn improved mappings, yet their black-box prediction mechanism inherently limits their ability to provide chemically-grounded rationales. In contrast, LLM-based methods rely on supervised fine-tuning (SFT) to enhance retrosynthetic capabilities, either from product-reactant SMILES pairs or structured rationales distilled from closed-source LLMs (Li et al., 2025). Despite enriching outputs or improving reactant prediction, this training signal often encourages imitation of fixed answer formats and memorized reaction patterns (Zhang et al., 2024; Zhao et al., 2024). This reliance limits the model's capacity to understand underlying chemical principles and perform logical reasoning. This paradigm faces three key limitations: (1) Existing work typically outputs only reactant SMILES, with no transparency in their internal reasoning processes and limited ability to provide natural language explanations grounded in chemical logic. (2) The pure pattern-matching training mechanism significantly constrains the model's capacity for effective logical decision-making. (3) The prevailing chemical-data supervision provides limited signal for learning flexible chemical reasoning beyond direct mapping. These limitations lead to a lack of reliable bases for the predictions in chemical principles, which undermines chemists' trust in practical applications.

Given these challenges, a reasoning-driven paradigm aligns more closely with the intrinsic nature of retrosynthetic analy-

sis than simplistic pattern matching, which chemical experts view as an iterative and logical reasoning process. Notably, recent breakthroughs in other domains (Wang et al., 2025; Xie et al., 2025; Chen et al., 2025) have demonstrated the potential of LLMs to address complex specialized problems through specialized-knowledge-based reasoning, enhanced by reinforcement learning. Therefore, we focus on *chemical knowledge-based retrosynthetic reasoning* by LLMs to generate reactants along with an explainable reasoning process, ensuring interpretable and transparent retrosynthesis prediction. To induce reasoning, it is crucial to select an appropriate learning paradigm equipped with effective chemical-data signals. Supervised Fine-Tuning (SFT) encourages models towards replicating prevalent reaction patterns observed in training data, rather than engaging in reasoning grounded in underlying chemical principles. In contrast, Reinforcement Learning (RL) approaches, which optimize the models' behavioral policy through reward feedback, offer a theoretically viable path to incentivize the complex and logical reasoning abilities for chemical tasks. Therefore, we aim to explore pure RL to induce flexible retrosynthetic reasoning without relying on SFT-based pattern imitation.

However, directly applying RL to incentivize models for retrosynthetic reasoning faces two critical challenges: (1) Domain Knowledge Disparity. Retrosynthesis demands not only logical reasoning but also mastery of specialized chemical knowledge. Pre-trained LLMs fail to adequately internalize and apply specific chemical principles when reasoning solely based on molecular SMILES. (2) The "Cold Start" in Exploration. For complex chemical reasoning tasks, LLMs struggle to identify valid solution paths when reasoning solely from molecular SMILES without domain guidance, resulting in sparse feedback that hinders strategy optimization. To bridge this gap, we propose a novel paradigm (in Figure 1) that synergistically combines (1) the distilled knowledge priors from specialized models with (2) the deep logical reasoning of LLMs. This domain-specific knowledge is more task-grounded and less hallucination-prone than free-form rationales distilled from closed-source LLMs, providing more reliable signals for pure RL.

Building upon these insights, we present an **Exp**lainable and Coop**er**ative **retro**synthesis framework, **Retro-Expert**, which integrates natural language-based chemical reasoning with model-agnostic compatibility. The framework contains three synergistic core components: (1) **Chemical Decision Space Construction.** Leveraging specialized models (e.g., reaction type classifiers, reaction center localization and reactant generators) to construct a high-quality chemical decision space, which provides "knowledge anchors" for the LLM's subsequent chemical reasoning. (2) **Collaborative Reasoning Engine.** The LLM performs critical-generative reasoning upon the chemical decision space. It dynamically selects the valid knowledge prior or generates

a novel one, constructing an interpretable, natural language-based reasoning process along with the reactant prediction. (3) **Knowledge-Grounded Policy Optimization (KGPO).** Retro-Expert leverages KGPO to optimize the LLM's reasoning strategy, employing a novel reward model (Chem-RM) informed by chemical tools to guide the model toward learning an optimal and trustworthy reasoning path.

Our contributions are summarized as follows:

1. We introduce **Retro-Expert**, an interpretable retrosynthesis framework that generates natural-language reasoning processes grounded in chemical logic.

2. We propose a collaborative reasoning paradigm built upon the synergy between the LLM and specialized models, enabling pure RL optimization with chemical knowledge signals. It improves reasoning accuracy and reliability through KGPO with Chem-RM.

3. Systematic experiments validate the advantages of Retro-Expert's LLM-specialized model collaborative reasoning, while exhibiting strong generalization and efficient extensibility to multi-step retrosynthesis.

**Conflict of Interest Disclosure.** The authors declare that there are no conflicts of interest.

## 2. Related Work

**Retrosynthesis Prediction.** Existing methods are broadly categorized into two modeling paradigms: specialized machine learning (ML) models and large language model (LLM)-based methods. *ML-based approaches* model chemical molecules from different perspectives (e.g., SMILES strings, molecular graphs) to learn the mapping from product to reactant molecules (Sacha et al., 2021; Chen & Jung, 2021; Somnath et al., 2021; Zhong et al., 2022). Although these methods demonstrate significant potential through precise molecular mappings, their black-box decision-making mechanisms inherently restrict the model's interpretability. *LLM-based approaches* aim to enhance the comprehension and memorization capabilities of LLMs for chemical tasks. Some efforts tailor LLMs to retrosynthesis via SFT (Zhang et al., 2024; Zhao et al., 2024; Yang et al., 2025b). Limited by SFT's memorization-centric fit to paired data, these models receive weak reasoning signals and tend to replicate memorized reaction patterns, producing only predicted reactants or options. Other work integrates the LLM with external tools to improve prediction accuracy (Bran et al., 2024; Liu et al., 2024; Bran et al., 2025). They primarily adopt or score given routes based on predefined metrics. While showing LLMs' capacity for route assessment, such strategies provide limited engagement in the predictive process and thus fall short of producing interpretable reasoning pathways that reveal underlying reaction mechanisms.

**Large Language Model Reasoning.** In recent years, the profound reasoning capabilities of large language models (LLMs) have been developed to solve specialized scientific problems (Su et al., 2025; Tang et al., 2025; Putri et al., 2025; Pan et al., 2025). While employing SFT (Zhou et al., 2023; Huang et al., 2025), the model primarily learns to replicate common reasoning patterns from curated datasets, restricting its effectiveness on more complex tasks. In contrast, reinforcement learning (RL)-based reasoning models recently achieve significant progress (Zhou et al., 2024; Guo et al., 2025; Feng et al., 2026). In the chemistry field, some studies have assessed the performance of LLMs on chemical reasoning tasks (Guo et al., 2023; Feng et al., 2024). These models predominantly focus on enhancing performance through SFT paradigm. However, there are few studies that incorporate expert-level reasoning strategies to tackle complex chemical reasoning tasks.

## 3. Methodology

### 3.1. Overview

Retro-Expert aims to enhance prediction accuracy while generating human-understandable reasoning processes. As illustrated in Figure 2, Retro-Expert is built upon three core modules. The process starts with Chemical Decision Space Construction, where specialized models analyze the target product and construct an anchored chemical decision space. This foundation activates the Collaborative Reasoning Engine, where the LLM performs chemical reasoning by analyzing molecular features, critically evaluating candidates, and articulating its multi-step logic into a complete natural language chain. Finally, Knowledge-Grounded Decision Policy Optimization employs RL to refine the LLM's critical decision-making strategy, using a novel reward model to guide it towards an optimal and trustworthy reasoning path.

### 3.2. Problem Formulation

The objective of the retrosynthesis task $T_{\text{retro}}$ is to predict the set of reactants $\{M_r^i\}_{i=1}^C (C \geq 1)$ corresponding to a target product $M_p$ with structural attributes $a_p$. We define an environment $\mathcal{E}$ that contains $s$ specialized models $\mathcal{M} = \{m_1, m_2, \ldots, m_s\}$, each model $m_i$ is responsible for generating Top-$N$ predictions as a candidate set $c_i$ corresponding to a distinct dimension of reaction-related information (e.g., reaction type, reaction center), defined as $c_i = \{c_i^n\}_{n=1}^N, c_i^n \sim p(m_i|M_p; \theta_i)$, where $\theta_i$ denotes the parameters of model $m_i$, and $c_i^n$ represents the $n$-th candidate result. The chemical decision space $\mathcal{T}$ is defined as the Cartesian product of candidate sets: $\mathcal{T} = (c_1, c_2, \ldots, c_s)$.

Upon this space, an LLM $M_{\text{LLM}}$ parameterized as a policy $\pi_\theta$ interacts with $\mathcal{E}$, analyzes the product structural attributes $a_p'$, selectively and critically evaluates the candidate set

$c_i$, and infers the correct answer $c_i'$ for the $i$-th reaction-related information. Subsequently, the LLM derives the final predicted set of reactants $a_r'$. This generates a reasoning path $\tau = (a_p', c_1', c_2', \ldots, c_{s'}', a_r')$ with $s' \leq s$, alongside a natural-language explanation $R$. The core objective of Retro-Expert is to learn an optimal policy $\pi^*$ that generates the optimal path $\tau^*$ with the highest reward:

$$\pi^* = \arg \max_{\pi_\theta} \mathbb{E}_{\tau \sim \pi_\theta} \left[ \text{RM}(\tau) \right], \qquad (1)$$

$$\tau^* = \arg \max_{\tau \sim \pi^*} \text{RM}(\tau) \qquad (2)$$

Here, $\text{RM}(\cdot)$ is a reward model evaluating the quality of the reasoning path. This framework improves the accuracy of explainable reasoning by jointly optimizing reactant prediction and the overall path-a critical enhancement not addressed in prior work.

### 3.3. Chemical Decision Space Construction

Due to the lack of domain-specific knowledge and experience, general LLMs face challenges in retrosynthesis when relying solely on molecular SMILES. Existing methods rely on SFT on large structured chain-of-thought (CoT) data for effective RL, which promotes rigid pattern replication rather than genuine chemical reasoning (Su et al., 2026). Meanwhile, directly applying RL often fails to efficiently explore feasible reasoning paths due to sparse reward signals, making it difficult for training to progress. Therefore, effective chemical knowledge signals are needed to guide LLMs toward more efficient exploration.

To overcome these challenges, we propose the chemical decision space, which provides LLMs with chemically meaningful starting points for reasoning. This design transforms chemical data from mapping-oriented supervision into reasoning-oriented guidance, enabling the model to explore feasible reaction pathways more efficiently while maintaining reasoning flexibility. Specifically, for a given target product $M_p$, we explicitly incorporate reaction-relevant information, such as reaction type and center, as chemically meaningful reasoning cues. We then invoke a suite of specialized models optimized for high-recall, each dedicated to a chemical perspective, to generate a corresponding set of plausible candidates. These candidate sets (e.g., reaction type, reaction center) are then integrated to form a multi-dimensional decision space. This space provides well-founded "knowledge anchors" that ground the LLM's subsequent reasoning, enabling knowledge-aware strategic planning and efficient exploration.

### 3.4. Collaborative Reasoning Engine

Building upon the chemical decision space, the Collaborative Reasoning Engine acts as the framework's cognitive hub, bridging specialized models with the LLM. It shifts the

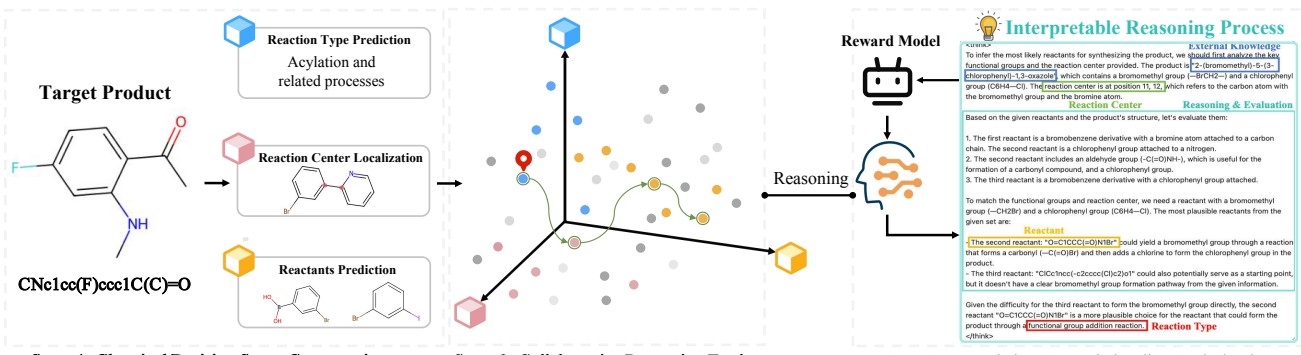

*Figure 2.* Overview of the Retro-Expert. (1) Decision Space Construction: Specialized models first analyze the target product to construct a high-dimensional chemical decision space composed of candidate pathways. (2) LLM-driven Navigation: Then, the LLM acts as a reasoning engine, strategically navigating the space via critical-generative reasoning. It can either select the best candidate pathway or generate a novel one if all provided options are deemed inadequate. (3) Policy Optimization: Finally, this navigation policy is optimized via Knowledge-Grounded Policy Optimization (KGPO) with a novel chemically grounded reward model, ensuring reasoning that is both accurate and chemically sound. The details of KGPO are visualized in Figure 4 in the Appendix.

LLM's role from a mere predictor or tool-caller to an active reasoning agent that autonomously navigates the retrosynthetic path, moving from "***directly predicting reactants***" to "***logical deduction***" toward the final conclusion.

The collaborative mechanism is centered on ***LLM-driven Critical–Generative Reasoning***. Within the chemical decision space, the LLM operates as the active chemical reasoner. In contrast to sequential subtask execution, the LLM autonomously performs logical reasoning by selectively utilizing the provided candidate sets, constructing a logically coherent retrosynthetic pathway. Specifically, the LLM's autonomous reasoning is a dynamic interplay of two core capabilities: ***critical analysis*** and ***generative decision-making***. At each reasoning step, the LLM first rigorously evaluates one candidate set within the provided context. This critical analysis, which ensures a comprehensive exploration of the problem space, directly informs the subsequent decision-making process, which manifests in one of two actions: (1) **Selection**: The LLM identifies and selects the most plausible candidate from the provided options. (2) **Rejection & Generation**: If its analysis concludes that no provided candidate is satisfactory, the LLM leverages its internal knowledge and reasoning context to generate a novel, self-consistent solution. These capabilities are learned by the LLM and manifested through autonomous reasoning, rather than hard-coded rules or prompt-specified procedures. This adaptive reasoning strategy, by mirroring human expert intuition, enables context-aware decisions that improve both the accuracy and explainability of the final prediction.

### 3.5. Knowledge-Grounded Policy Optimization

We introduce a novel reinforcement learning framework to optimize the LLM's reasoning policy guided by chemically grounded knowledge (see Figure 4 in the Appendix).

**Reward Modeling.** In scientific tasks, traditional reward modeling faces two main challenges. Training-based process-level reward models suffer from limited generalization and inherent hallucinations (Leng et al., 2025). In contrast, rule-based rewards that rely solely on final-answer supervision fail to provide granular learning signals for the reasoning process (Shao et al., 2024). Importantly, molecular properties and reaction behaviors follow objective principles, which can be evaluated using chemical tools.

Built upon it, we introduce Chem-RM ($r_\phi$), a training-free reward model that uses chemical toolkits to provide dense and accurate reward signals grounded in intrinsic chemical knowledge. The core objective is to incentivize the LLM to learn a chemically-grounded reasoning policy that prioritizes the logical validity of the full pathway. Specifically, given a query $q$ and a sampled decision path $y$, $r_\phi$ outputs a composite reward from three distinct modules:

$$r_\phi(q, y) = \lambda_1 r_{\text{struct}}(q, y) + \lambda_2 r_{\text{react}}(q, y) + \lambda_3 r_{\text{base}}(q, y) \quad (3)$$

The structure-related module $r_{\text{struct}}(q, y)$ evaluates the model's accuracy in analyzing the product's structural features. It employs RDKit to identify and compare molecular substructures, calculated as $|S_{\text{true}} \cap S_{\text{pred}}| / (|S_{\text{true}}| + \varepsilon)$, where $S_{\text{true}}$ denotes the set of true structural features of the product (e.g., functional groups, aromaticity), $S_{\text{pred}}$ represents the set of structural features in the decision path.

The reaction-related module $r_{\text{react}}(q, y)$ is designed to encourage the model to critically leverage predicted candidates (e.g., reaction type, center) upon the chemical decision space, and the reward is computed as:

$$\sum_{i=1}^{s} \left[ \mathbb{I}(c_i^{gt} \in y) + \mathbb{I}(c_i^{pred} \neq c_i^{gt}) \cdot \mathbb{I}((\neg c_i^{pred}) \in y) \right] \quad (4)$$

where $c_i^{gt}$ and $c_i^{pred}$ denote the ground-truth and the corre-

*Table 1.* Comparison with LLM-based models on USPTO-50K (Schneider et al., 2016). Space Setting indicates whether the model is provided with Top-4 reactant candidates and additional domain knowledge anchors (i.e., reaction types, reaction center) from specialized models. General LLMs excel at logical reasoning but lack chemical knowledge. When equipped with a chemical decision space, their performance improves substantially. Chemical LLMs tend to replicate memorized reaction patterns rather than perform genuine reasoning.

| Model | Space Setting | Top-1 (%)↑ | BLEU↑ | Dis↓ | Val.↑ | MACCS↑ | RDK↑ | Morgan↑ |
|---|---|---|---|---|---|---|---|---|
| *LLM-based chemical models* | | | | | | | | |
| ChemFormer (Irwin et al., 2022) | ✗ | 27.30 | 0.769 | 14.77 | 0.952 | 0.782 | 0.690 | 0.647 |
| T5Chem (Lu & Zhang, 2022) | ✗ | 43.64 | 0.972 | 8.42 | 0.982 | 0.956 | 0.974 | 0.925 |
| | ✓ | 27.88 | 0.963 | 13.85 | 0.977 | 0.937 | 0.956 | 0.883 |
| nach0 (Livne et al., 2024) | ✗ | 30.52 | 0.976 | 11.91 | 0.990 | 0.947 | 0.977 | 0.905 |
| | ✓ | 14.60 | 0.962 | 16.10 | 0.981 | 0.891 | 0.942 | 0.805 |
| ChemDFM (Zhao et al., 2024) | ✗ | 6.69 | 0.743 | 16.62 | 0.874 | 0.714 | 0.807 | 0.617 |
| | ✓ | 13.64 | 0.898 | 11.73 | 0.841 | 0.892 | 0.928 | 0.819 |
| ChemLLM (Zhang et al., 2024) | ✓ | 12.83 | 0.869 | 10.34 | 0.979 | 0.851 | 0.849 | 0.773 |
| ChemCrow (+nach0) (Bran et al., 2024) | ✗ | 30.52 | 0.976 | 11.91 | 0.990 | 0.947 | 0.977 | 0.905 |
| | ✓ | 14.60 | 0.962 | 16.10 | 0.981 | 0.891 | 0.942 | 0.805 |
| *LLM-based general models* | | | | | | | | |
| InterLM2-Chat-7B | ✗ | 0.73 | 0.743 | 19.05 | 0.706 | 0.781 | 0.764 | 0.589 |
| | ✓ | 24.31 | 0.957 | 11.03 | 0.895 | 0.945 | 0.952 | 0.909 |
| GPT-3.5 | ✗ | 1.02 | 0.836 | 17.27 | 0.752 | 0.788 | 0.803 | 0.595 |
| | ✓ | 32.93 | 0.969 | 7.76 | 0.991 | 0.952 | 0.968 | 0.913 |
| GPT-4o | ✗ | 0.30 | 0.839 | 17.68 | 0.751 | 0.743 | 0.646 | 0.493 |
| | ✓ | 32.85 | 0.970 | 7.72 | 0.990 | 0.953 | 0.970 | 0.914 |
| Gemini-2.5-preview-05-20 | ✗ | 4.19 | 0.790 | 17.20 | 0.864 | 0.765 | 0.720 | 0.572 |
| | ✓ | 39.02 | 0.986 | 6.63 | 0.961 | 0.973 | 0.989 | 0.944 |
| Gemini-2.5-Pro | ✗ | 6.83 | 0.845 | 15.96 | 0.879 | 0.786 | 0.789 | 0.604 |
| | ✓ | 50.61 | 0.988 | 5.58 | 0.983 | 0.970 | 0.985 | 0.938 |
| Claude-3-7-sonnet | ✗ | 5.73 | 0.858 | 16.47 | 0.865 | 0.808 | 0.768 | 0.577 |
| | ✓ | 55.70 | 0.990 | 4.87 | 0.987 | 0.974 | 0.983 | 0.956 |
| Qwen2.5-7B-Instruct (Zero-shot) | ✗ | 0.02 | 0.670 | 30.80 | 0.668 | 0.522 | 0.396 | 0.289 |
| | ✓ | 37.55 | 0.971 | 7.64 | 0.992 | 0.961 | 0.973 | 0.922 |
| Qwen2.5-7B-Instruct (Few-shot) | ✗ | 2.14 | 0.724 | 25.86 | 0.756 | 0.645 | 0.581 | 0.491 |
| | ✓ | 40.66 | 0.977 | 5.92 | 0.992 | 0.966 | 0.974 | 0.928 |
| Qwen2.5-7B-Instruct (SFT) | ✗ | 43.62 | 0.980 | 5.36 | 0.993 | 0.973 | 0.979 | 0.940 |
| | ✓ | 58.92 | 0.985 | 4.91 | 0.995 | 0.979 | 0.980 | 0.958 |
| **Ours** | ✓ | **75.69** | **0.998** | **2.01** | **0.997** | **0.993** | **0.996** | **0.991** |

sponding predicted plausible candidate for the $i$-th reaction-related information. $\mathbb{I}(\cdot)$ is the indicator function, assigning 1 if the condition is satisfied, 0 otherwise. The first term promotes the presence of correct candidate information, covering both the preservation of correct predictions and the correction of incorrect ones. The second term promotes the rejection of incorrect predicted candidates. These behaviors are induced by LLM's reasoning via RL, rather than from hard-coded rules or prompt templates.

The basic module's reward is defined as $r_{\text{base}}(y) = r_{\text{ans}}(q, y) + \alpha r_{\text{fmt}}(y)$, where $r_{\text{ans}}$ serves as a binary indicator of the correctness of the final predicted reactants, $r_{\text{fmt}}$ aims to ensure the required output format.

**Optimization.** Building upon the proposed Chem-RM, we aim to optimize the LLM's reasoning policy by maximizing expected reward $r_\phi(q, y)$, formally defined as:

$$\max_{\pi_\theta} \mathcal{J}_{KGPO}(\theta) = \mathbb{E}_{q \sim \mathcal{D}, y \sim \pi_\theta(\cdot|q;\mathcal{E})} [r_\phi(q, y)] \\ - \beta \mathbb{D}_{kl} [\pi_\theta(y \mid q; \mathcal{E}) \mid \pi_{\text{ref}}(y \mid q; \mathcal{E})], \quad (5)$$

In this objective function, $\pi_\theta$ represents the policy model being optimized, while $\pi_{ref}$ is a reference model used to regularize the policy update, with their divergence measured by $\mathbb{D}_{kl}$. To optimize this objective, we build upon the Group Relative Policy Optimization (GRPO) algorithm (Shao et al., 2024). However, the standard GRPO computes advantages

via linear group-wise normalization, which compresses the relative advantage of high-value reasoning paths that correct erroneous candidates, hindering the LLM from learning critical corrective behaviors.

To address this limitation, we propose Knowledge-Grounded Policy Optimization (KGPO), which introduces a non-linear rectification to the advantage estimation, explicitly amplifying learning signals from critical correction reasoning. For the $i$-th decision path in a sampled group $G$, the rectified advantage is computed as:

$$\hat{A}_i = \frac{r_\phi(q, y_i) - \mu_G}{\sigma_G + \epsilon} \cdot \left(1 + \gamma \frac{r_{\text{crit}}(y_i)}{\max_j r_{\text{crit}}(y_j) + \epsilon}\right) \quad (6)$$

where $\mu_G$ and $\sigma_G$ denote the group-wise mean and standard deviation of rewards, and $\epsilon$ ensures numerical stability. $r_{\text{crit}}(y)$ represents the reward for critical corrections, computed as $r_{\text{crit}}(y) = \sum_{i=1}^s \mathbb{I}(c_i^{\text{pred}} \neq c_i^{\text{gt}}) \cdot \mathbb{I}((\neg c_i^{\text{pred}}) \in y)$. The policy parameters are then updated according to:

$$\theta \leftarrow \theta + \eta \left(\frac{1}{G} \sum_{i=1}^G \hat{A}_i \nabla_\theta \log \pi_\theta(y_i|q; \mathcal{E}) - \beta \nabla_\theta \mathbb{D}_{\text{kl}}\right) \quad (7)$$

By integrating KGPO with Chem-RM, Retro-Expert shifts its focus from "***accuracy-centric prediction***" to the "***coherent and interpretable reasoning chains generation***". Furthermore, KGPO enhances the LLM's critical decision-

making within the decision space, steering the policy toward correct conclusions via chemically valid reasoning.

# 4. Experiments

## 4.1. Experimental Setup

**Dataset.** We conduct experiments using two benchmark datasets: USPTO-50K (Schneider et al., 2016) and USPTO-FULL (Dai et al., 2019), which contain 50k and 1 million atom-mapped reaction records, respectively. We adopt the same training/validation/test splits (8:1:1) as prior work (Dai et al., 2019). Following previous methods (Somnath et al., 2021), we canonicalize the product SMILES and reassign the atom-mapping to the corresponding reactant SMILES based on the canonical ordering.

**Baselines.** We compare Retro-Expert against various strong baselines, categorized into two primary classes: (1) non-LLM-based approaches, including MHNReact (Seidl et al., 2022), LocalRetro (Chen & Jung, 2021), GLN (Dai et al., 2019), GraphRetro (Somnath et al., 2021), Retro-Prime (Wang et al., 2021), Graph2Edits (Zhong et al., 2023), Retroformer (Yao et al., 2024b), UAlign (Zeng et al., 2024); (2) LLM-based methods, including chemistry-specialized LLMs and general LLMs.

**Evaluation Metrics.** Inspired by (Zhao et al., 2024), we adopt two complementary sets of metrics to evaluate the performance between the predicted and golden standardized SMILES. The first set focuses on the *textual similarity*, using Top-1 accuracy, BLEU, and Levenshtein distance. The second set evaluates the *chemical similarity*, encompassing the validity of the generated SMILES and fingerprint Tanimoto similarity (i.e., MACCS, RDK, Morgan). Additionally, we compare the performance of Retro-Expert against retrosynthesis small models using Top-K accuracy.

**Implementation Details.** In training, we employ specialized models for space construction, i.e., T5Chem (Lu & Zhang, 2022) for reaction type prediction, and GraphRetro (Somnath et al., 2021) for reaction center localization and reactant prediction. During inference, *ANY* retrosynthetic models can be used to provide reactant candidates in a plug-and-play way, without retraining the LLM. To balance accuracy and optimization efficiency, we use Top-4 candidate predictions from each model. We utilize Qwen2.5-7B-Instruct (Yang et al., 2025a) as the LLM and train it using 9k and 80k samples from USPTO-50K (Schneider et al., 2016) and USPTO-FULL (Dai et al., 2019) datasets, respectively. During training, to prevent reward hacking, we distribute the correct predictions from the specialized model among the Top-4 candidate positions in a 5:3:2 ratio. This strategy prevents the model from developing a positional bias (e.g., always selecting the first candidate). The reward weighting coefficients are set as $\lambda_1 = 1.5$,

*Table 2.* Comparison with non-LLM-based models on USPTO-50K (Schneider et al., 2016). Ours indicates the same LLM that reasons over a decision space constructed by the corresponding retrosynthesis model.

| Method | Top-K accuracy (%) | | | | Interpre-table |
|---|---|---|---|---|---|
| | 1 | 3 | 5 | 10 | |
| MHNReact | 51.8 | 74.6 | 81.2 | 88.1 | ✗ |
| Retroformer | 63.5 | 82.1 | 86.3 | 90.2 | ✗ |
| LocalRetro | 63.9 | 86.8 | 92.4 | 96.3 | ✗ |
| GraphRetro | 63.9 | 81.5 | 85.2 | 88.1 | ✗ |
| GLN | 64.2 | 79.1 | 85.2 | 90.0 | ✗ |
| UAlign | 66.2 | 86.9 | 91.9 | 95.1 | ✗ |
| Graph2Edits | 67.2 | 87.5 | 91.5 | 93.8 | ✗ |
| Ours (MHNReact) | 60.9 | 77.8 | 84.6 | 89.3 | ✓ |
| Ours (Retroformer) | 68.7 | 83.5 | 86.3 | 90.4 | ✓ |
| Ours (LocalRetro) | 70.7 | 90.4 | **95.0** | **96.7** | ✓ |
| Ours (GraphRetro) | 70.1 | 83.0 | 86.4 | 88.2 | ✓ |
| Ours (GLN) | 69.8 | 81.7 | 87.2 | 90.4 | ✓ |
| Ours (UAlign) | 73.4 | 90.6 | 94.4 | 95.6 | ✓ |
| Ours (Graph2Edits) | **75.7** | **90.8** | 93.1 | 94.1 | ✓ |

*Table 3.* Comparison of different retrosynthesis methods on the USPTO-FULL (Dai et al., 2019) dataset.

| Method | Top-1 Acc | Top-3 Acc |
|---|---|---|
| LocalRetro (Chen & Jung, 2021) | 39.1 | 53.3 |
| GLN (Dai et al., 2019) | 39.3 | – |
| Graph2Edits (Zhong et al., 2023) | 44.0 | 60.9 |
| GTA (Seo et al., 2021) | 46.6 | – |
| EditRetro (Han et al., 2024) | 47.1 | 61.2 |
| NAG2G (Yao et al., 2024a) | 49.7 | 64.6 |
| **Ours** | **58.4** | **67.3** |

$\lambda_2 = 2.0$, and $\lambda_3 = 1.0$.

## 4.2. Comparison with State of the Art

***Superiority over LLM-based Approaches.*** As detailed in Table 1, Retro-Expert significantly outperforms both chemistry-focused foundation models and general LLMs, including even stronger and larger LLMs. Moreover, Retro-Expert achieves a Top-1 accuracy improvement of over 16.77% over alternative training strategies (i.e., zero-shot, few-shot, SFT) under the same backbone, suggesting that the gains cannot be explained by simply retrieving memorized priors alone. This demonstrates that our knowledge-grounded reinforcement learning approach effectively enhances the model's reasoning capability. In contrast, the chemical LLMs trained via SFT tend to replicate memorized reaction patterns for prediction. This strategy exhibits suboptimal performance for retrosynthesis prediction.

***Superiority over Non-LLM-based Approaches.*** Beyond outperforming LLMs, Retro-Expert also demonstrates clear advantages over non-LLM-based models (Table 2). When using different retrosynthesis models to construct the reactant-

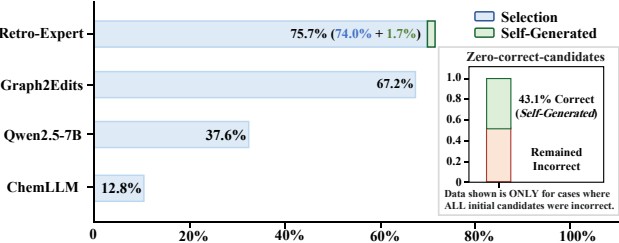

*Figure 3.* Comparison of correct predictions distributions across different models and Success rate of Retro-Expert's self-correction mechanism.

*Table 4.* Top-1 accuracy (%) of different models on the out-of-distribution ChemBench benchmark (Zhang et al., 2024).

| Method | Top-1 Acc (%) |
|---|---|
| ChemLLM (Zhang et al., 2024) | 15.33 |
| Qwen2.5-7B-Instruct (Yang et al., 2025a) | 29.33 |
| DeepSeek-R1 (Guo et al., 2025) | 32.54 |
| **Ours** | **65.67** |

*Table 5.* Human evaluation of reasoning correctness on 500 randomly selected samples.

| Answer | Reasoning Process | Coverage (%) |
|---|---|---|
| Correct | Correct | 82.85 |
| Correct | Partially Correct | 17.15 |
| Wrong | Correct for a Plausible Alternative | 33.60 |
| Wrong | Partially Correct & Incorrect | 66.40 |

related decision space, Retro-Expert achieves higher Top-K accuracy than the underlying small models. Moreover, our method exhibits model-agnostic compatibility, enabling seamless integration with various specialized models at no cost (no retraining required). The performance gains scale proportionally with the baseline model's own accuracy. These results validate that Retro-Expert effectively reasons over the provided candidate information to optimize its final decision. This collaborative mechanism serves a dual purpose: it boosts the predictive performance of the entire system while making the final choice fully interpretable.

Beyond the USPTO-50K dataset, we further evaluate the performance on the larger and more diverse USPTO-FULL dataset. As shown in Table 3, our method achieves superior performance to existing approaches. This demonstrates that our method is capable of handling more complex and diverse reaction scenarios.

***Incentivized Capability for Self-Reflection.*** A particularly compelling finding is the incentivized capability of Retro-Expert for self-reflection and reasoning, which allows the LLM to overcome the limitations of the input specialized models. The framework does not uncritically accept the candidates provided by specialized models. Instead, it exhibits a capacity for critical analysis and self-correction induced by KGPO. As illustrated in Figure 3, when a specialized model fails to provide valid reactant candidates (e.g., they are entirely incorrect or no candidate), Retro-Expert may reject the provided candidates and generate a new correct answer. A statistical analysis of such cases reveals that this self-reflection mechanism achieved a remarkable 43.1% success rate, showing that it is not merely a selection model but a generative model that can create new answers. The detailed analyses of model behaviors toward candidates (e.g., select, reject, self-correct) are provided in Appendix A.

***Superior Generalization on Out-of-Distribution Data.*** We evaluate Retro-Expert's generalization on the out-of-distribution (OOD) ChemBench benchmark (Zhang et al., 2024), where 94.66% of the 300 reaction pathways are absent from USPTO-50K, reflecting real-world discovery con-

ditions. To ensure fairness, all models shared an identical decision space formulated as a four-option choice task (one ground truth and three distractors). As shown in Table 4, general LLMs achieve accuracy around 30%. In contrast, Retro-Expert attains 65.67% accuracy. This improved generalization is enabled by KGPO, which ingrains a transferable, chemistry-principled reasoning policy.

### 4.3. Reasoning Process Evaluation

We evaluate the generated reasoning from two complementary aspects, **Reasoning Reliability** and **Explanation Quality**. The former assesses whether the reasoning faithfully supports the final prediction, while the latter evaluates whether the generated explanation is chemically credible and understandable to experts.

#### 4.3.1. REASONING RELIABILITY

We assess reasoning reliability along two dimensions, **Reasoning Correctness** and **Decision Consistency**. For reasoning correctness, we randomly selected 500 samples and asked three chemistry PhDs in organic synthesis to rigorously evaluate the model's reasoning, judging whether the reasoning is correct. For decision consistency, we examine the correctness of intermediate decisions (i.e., reaction type, reaction center, reactants) in the model's reasoning on the test set. The details of the reasoning correctness annotation are provided in Appendix B.

***Retro-Expert Achieves Reliable Reasoning Coupled with Final Predictions.*** As shown in Table 5, the proportion of correct reasoning is higher for correct answers (82.85%) than for incorrect ones (33.60%). This strong alignment between reasoning correctness and answer correctness suggests that Retro-Expert's reasoning is closely coupled with its final prediction. In addition, Table 14 shows that correct answers are typically supported by fully correct intermediate

*Table 6.* Evaluation on reasoning process using GPT-4o and human expert assessment. MA: Mechanism Accuracy, FC: Factual Correctness, LC: Logic Consistency.

| Evaluator | Model | Metrics | | |
|-----------|-------|---------|---|---|
| | | MA | FC | LC |
| GPT-4o | Qwen2.5-7B-Instruct | 3.67 | 3.12 | 4.06 |
| | **Ours** | **4.41** | **4.17** | **4.33** |
| Human | Qwen2.5-7B-Instruct | 3.21 | 2.89 | 3.87 |
| | **Ours** | **4.35** | **3.98** | **4.34** |

decisions (64.06%), while wrong answers are more often accompanied by flawed decisions (17.35%). This indicates the correctness of intermediate decisions is well aligned with the final prediction. These results show that Retro-Expert's reasoning process causally contributes to final predictions, rather than merely generating post-hoc rationalizations.

#### 4.3.2. EXPLANATION QUALITY

Since explanatory capabilities are largely absent in specialized chemical models, we benchmarked Retro-Expert against general LLMs on the USPTO-50K test set. These baseline models were explicitly prompted to generate a step-by-step rationale. The evaluation combines GPT-4o automatic scoring with human assessment by three independent experts in synthetic organic chemistry.

We design three core metrics: (1) Mechanism Accuracy (MA): The alignment of the described reaction mechanism with established chemical principles; (2) Factual Correctness (FC): The accuracy of chemical knowledge within the reasoning process; and (3) Logical Consistency (LC): The logical coherence and soundness of the reasoning. Each metric was scored on a 1–5 scale (5 being the best), with definitions and detailed criteria provided in the Appendix B.

***Retro-Expert Delivers Credible Reasoning Superior to General LLMs.*** As detailed in Table 6, Retro-Expert achieves scores above 4.0 in the automated GPT-4o assessment, significantly outperforming the general LLM baselines. This strong performance was largely corroborated by human experts, who awarded high scores for MA and LC, confirming that Retro-Expert generates clear, coherent, and mechanistically plausible reasoning pathways. These demonstrated capabilities are crucial for enhancing the transparency and interpretability of the prediction process, fostering greater trust in its real-world application.

### 4.4. Ablation Study

We conducted a series of ablation studies to validate the core architectural choices of Retro-Expert. Additional studies are provided in the Appendix E.

***Efficacy of the Reward Model.*** To substantiate the efficacy of our reward model, we conduct a component-wise

*Table 7.* Ablation study on the components of our reward model.

| Reward Model | Reasoning Result | Reasoning Process | | |
|--------------|------------------|-------------------|---|---|
| | Top-1 Acc (%) | MA | FC | LC |
| $r_{base}$ | 66.8 | 3.37 | 3.65 | 3.49 |
| $r_{base} + r_{react}$ | 72.3 | 4.35 | 4.01 | 4.28 |
| Ours | **75.7** | **4.41** | **4.17** | **4.33** |

*Table 8.* Performance comparison using different training policies. Top-1 accuracy quantifies reasoning correctness, and MA, FC, and LC assess the reasoning process.

| Setting | Reasoning Result | Reasoning Process | | |
|---------|------------------|-------------------|---|---|
| | Top-1 Acc (%) | MA | FC | LC |
| SFT (with CoT) | 48.6 | 2.75 | 2.12 | 2.84 |
| RL (GRPO) | 66.8 | 3.37 | 3.65 | 3.49 |
| RL (KGPO) | **75.7** | **4.41** | **4.17** | **4.33** |

ablation study within the RL framework using KGPO. As shown in Table 7, relying solely on outcome-based rewards yields suboptimal performance in both accuracy (66.8%) and reasoning quality. When gradually incorporating the reaction-related module $r_{react}$ and structure-related module $r_{struct}$, predictive accuracy increases from 72.3% to 75.7%, paralleled by consistent enhancements in reasoning quality. These results highlight that dense, multi-dimensional supervision is indispensable for fostering accurate and chemically sound reasoning paths.

***Effectiveness of KGPO for Superior Reasoning.*** To understand the effectiveness of KGPO in enhancing LLM reasoning, we train LLMs using different training strategies. As shown in Tables 1 and 8, the KGPO-trained LLM achieves superior performance compared to other baselines. Prompt-based strategies benefit from the provided candidate information, yet their performance remains constrained without policy optimization. SFT encourages pattern memorization for prediction. This limits the model's ability to flexibly perform logical reasoning, thereby creating performance bottlenecks in reasoning-intensive tasks such as retrosynthesis. While GRPO improves accuracy, it solely focuses on the final answer's correctness. This may lead the model to exploit logically flawed or chemically implausible shortcuts to arrive at the correct conclusion, resulting in suboptimal reasoning outcomes and paths. In contrast, our KGPO strategy effectively guides the model in utilizing the structured decision space, resulting in more accurate predictions. Furthermore, the strategy notably enhances the quality of the reasoning process, improving both the coherence and accuracy of the generated content.

### 4.5. Wet Lab Experiments

To validate Retro-Expert's practical utility, we extended beyond in-silico (dry-lab) tests to chemical wet-lab experi-

*Table 9.* Comparison of different multi-step retrosynthesis methods. All multi-step retrosynthesis models are run with a beam size of 50. More results are provided in the Appendix C.

| Method | Success Rate (%) | Top-1 Acc. (%) | Top-5 Acc. (%) | Inference Time /sample (s) |
|---|---|---|---|---|
| DFPN | 84.75 | 11 | 17 | 59.47 |
| MCTS | 97.14 | 17 | 46 | 51.73 |
| Retro* | 97.26 | 15 | 41 | 51.16 |
| Ours | **98.39** | **41** | **54** | **20.41** |

ments. We provide direct experimental evidence that Retro-Expert can successfully propose a feasible synthesis of a molecule that previously lacked any documented production path, and can discover a novel reaction route for an existing compound. The case selection criteria and experimental details are provided in Appendix D.

Our experiments successfully validate these two distinct types of chemical discovery. (1) Our Retro-Expert model predicted a new **Suzuki-Miyaura coupling** route to synthesize 3-(2-ethoxyphenyl)thiophene (CCOC1=CC=CC=C1C1=CSC=C1) via a reaction between 1-bromo-2-ethoxybenzene (CCOc1ccccc1Br) and thiophen-3-ylboronic acid (OB(O)c1ccsc1). Crucially, verification using the CAS SciFinder database confirmed that while the compound's structure was known, no synthesis pathway had ever been published. Our work thus constitutes its first-ever reported synthesis, successfully achieved with 79.3% yield. (2) For the well-established molecule 1-(4-ethoxyphenyl)ethanone (CCOC1=CC=C(C=C1)C(C)=O), our model identified a novel **Jones Oxidation** pathway. The target compound can be synthesized via 1-(4-ethoxyphenyl)ethanol (CCOc1ccc(C(C)O)cc1) using chromium trioxide in acidic conditions. This previously undocumented alternative was successfully executed in our chemical lab with a yield of 58.82%. Both cases are our model's Top-1 predictions, and the key chemical analyses in the reasoning were judged logically plausible by chemists. These outcomes offer compelling proof that Retro-Expert is an effective tool for practical chemical discovery.

### 4.6. Case Study: Making Reasoning Transparent

As visualized on the right side of Figure 2, Retro-Expert generates a human-readable rationale that articulates its critical analysis for a specific molecule. For this example, specialized models identify "functional group addition" as a probable reaction type and pinpoint positions 11 and 12 as potential reaction centers. The model's rationale then demonstrates its navigation of the chemical space. For instance, it correctly discards a candidate reactant, explicitly stating that it is ruled out due to the "absence of a clear bromomethyl group". This logical and verifiable narrative is essential for enhancing a chemical expert's trust and boosting the model's practical applicability.

### 4.7. Multi-Step Retrosynthesis Prediction

We extend Retro-Expert to multi-step retrosynthesis prediction based on the PaRoutes dataset (Genheden & Bjerrum, 2022). Rather than invoking the LLM at each node expansion as in MCTS routing, Retro-Expert decouples route search from reasoning. Specifically, the multi-step specialized model DMS-Model (Shee et al., 2025) first generates Top-4 complete routes through beam search. These routes are jointly provided to the LLM as a route-level decision space. The LLM then performs a single round of reasoning over the candidate routes. This design assigns efficient search to specialized models and reserves high-value chemical reasoning for the LLM.

***Retro-Expert Enables Efficient Multi-Step Planning.*** As shown in Table 9, Retro-Expert consistently outperforms existing methods across multiple metrics, achieving the highest prediction accuracy with the lowest inference time. These results underscore the potential of Retro-Expert for practical retrosynthesis prediction.

***Retro-Expert Shows Practical Value on Challenging Targets.*** We further evaluate Retro-Expert on challenging synthetic targets beyond standard benchmarks. For 3-oxa[3.1.1]propellane and 3-aza[3.1.1]propellane, the predicted routes are consistent with the reported synthetic strategies (Revie et al., 2026). These results suggest that Retro-Expert has the potential to identify novel synthetic transformations.

***Retro-Expert Goes Beyond Route Reranking through Step-Aware Reasoning.*** Beyond route reranking, Retro-Expert performs step-aware reasoning over the candidate routes. This enables our model to exploit partially correct local decisions across different routes and combine them into a better final route (e.g., using one branch from Route-A and another branch from Route-B to form the new route). It can also generate a new route conditioned on the candidate routes when it finds none of them satisfactory. The detailed examples are provided in Appendix C.

## 5. Conclusion

We introduced Retro-Expert, a novel retrosynthesis framework that performs chemical reasoning while producing human-readable reasoning processes. Our approach synergizes specialized models, which construct a chemical decision space, with an LLM that navigates it via KGPO, guided by a novel reward model providing chemically grounded signals for critical reasoning. Experiments show that Retro-Expert achieves competitive accuracy and strong generalization while allowing for plug-and-play modularity. By bridging the gap between AI prediction and chemists' workflows, Retro-Expert represents a significant step toward trustworthy, collaborative AI for chemical discovery.

## Acknowledgements

This work was supported by the National Science and Technology Major Project (2023ZD0120802).

## Impact Statement

This paper presents work whose goal is to advance the field of artificial intelligence for scientific discovery in chemistry. There are many potential societal consequences of our work, none which we feel must be specifically highlighted here.

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

# Appendix

This supplemental document provides details on our **Retro-Expert**. The organization is as follows:

- **Section A:** Describes the technical implementation of Retro-Expert.

- **Section B** Presents the reasoning process evaluation, covering (1) reasoning reliability evaluation through human-annotated reasoning correctness and decision consistency analysis, as well as (2) explanation quality evaluation based on evaluation metrics and human scoring procedures.

- **Section C:** Provides additional details and examples for multi-step retrosynthesis, showing how Retro-Expert performs step-aware reasoning beyond simple route reranking.

- **Section D:** Provides the details of our wet-lab experiments.

- **Section E:** Presents additional ablation studies, including the necessity of the chemical decision space, the mitigation of reward hacking, and the impact of varying the number of candidates on prediction performance.

- **Section F:** Visualizes representative reasoning examples to analyze the reasoning capabilities of Retro-Expert, including successful selection and generation cases, as well as failure cases.

- **Section G:** Presents all the prompts used in LLM-based baselines.

- **Section H:** Provides the prompts applied for all models when evaluated on the out-of-distribution (OOD) dataset.

- **Section I:** Discusses the limitations of Retro-Expert.

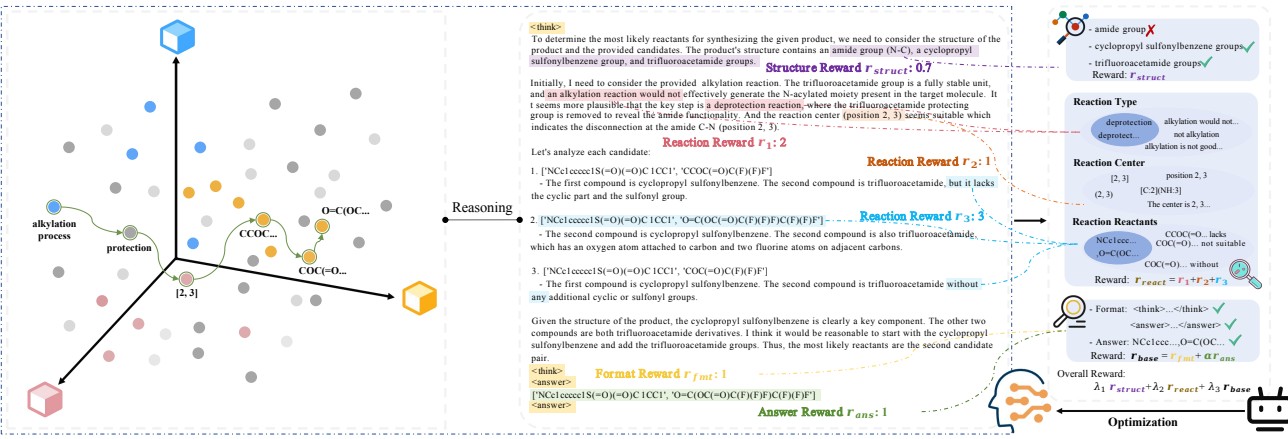

*Figure 4.* Visualization of the KGPO's optimization pipeline. Our Retro-Expert conducts case-by-case critical thinking and reasoning within the decision space, thereby generating an interpretable reasoning pathway along with the final answer. We further employ a novel reward model that evaluates the reasoning text. This reward mechanism guides the model in two ways: it encourages rejecting incorrect candidates (e.g., "alkylation would not" receives a reward of 1) while promoting inference of the correct reaction type (e.g., "a deprotection" receives a reward of 1).

## A. Details of KGPO Module

This section elaborates on the technical implementation details of our Retro-Expert. As depicted in Figure 4, we illustrate the optimization workflow of the KGPO module. This design underscores that the core of our KGPO is to promote critical thinking and reasoning grounded in chemical logic, rather than merely rewarding superficial correctness.

## A.1. Training Setting

**Training Samples.** To optimize training efficiency and enhance the model's ability to learn diverse chemical reaction principles, we did not utilize the entire training dataset directly. Instead, we meticulously curated a high-quality training subset. The original USPTO-50K dataset exhibited a significant long-tailed distribution of reaction templates and severe imbalance in sample quantities across different reaction types.

Our screening strategy was designed to mitigate these data skews. First, consistent with prior work, we employed chemical tools (i.e., RDKit) to extract reaction patterns from the training set using a subgraph pattern matching approach. All samples were partitioned into five tiers based on their template frequency. Subsequently, we performed stratified random sampling: samples were drawn from each frequency-based tier while ensuring that the final selected subset maintained a balanced distribution across reaction types. This process yielded a compact and balanced training set comprising 9k samples, providing a more robust data foundation for the model to learn reliable reasoning capabilities.

For the USPTO-FULL dataset, we applied a similar strategy to obtain a training subset. Due to the absence of ground-truth reaction type labels, we conducted stratified random sampling solely based on the frequency of extracted reaction patterns, resulting in a final training set of 80k samples.

**Training Configuration.** We employ reinforcement learning via GPRO (Generalized Reinforcement Learning with Policy Optimization) to optimize the reasoning strategy of the LLM (Qwen2.5-7B-Instruct). To ensure training stability while promoting deep deliberation and exploration, the core hyperparameters are configured as follows: train_epochs=2, max_completion_length=8192, temperature=0.7, $\beta$=0.001.

**Reward Model.** This reward model is designed to evaluate the complete decision-making process from the reasoning path to the final prediction.

$$r_\phi(q, y) = \lambda_1 r_{\text{struct}}(q, y) + \lambda_2 r_{\text{react}}(q, y) + \lambda_3 r_{\text{base}}(q, y) \tag{8}$$

The reaction-related module ($r_{react}$)'s reward comprises distinct intermediate rewards ($r_1$, $r_2$, $r_3$) corresponding to three sub-tasks: reaction type prediction, reaction center localization, and reactant prediction, respectively. Additionally, the reward of the basic module ($r_{base}$) consists of two components: $r_{\text{ans}}$ provides the final reward for the correctness of predicted reactants, while $r_{\text{fmt}}$ incentivizes strict adherence to the specified output structure: `<think>...</think><answer>...</answer>`. The corresponding weighting coefficients are set as $\lambda_1 = 1.5$, $\lambda_2 = 2.0$, and $\lambda_3 = 1.0$.

## A.2. KGPO-Induced LLM Behavior of Retro-Expert

**Candidate-Grounded Behavior Analysis.** To examine whether the policy learned by KGPO is actually working during inference, we further provide a fine-grained analysis of LLM behaviors in the reasoning process. Specifically, we inspect whether the model explicitly analyzes the provided candidates before making the final decision.

*Table 10.* Analysis of KGPO-induced LLM behaviors toward candidates in Retro-Expert's reasoning process.

| Candidate-Grounded Analysis | LLM Behavior | Behavior Coverage (%) | Correct Predictions via this Behavior (%) |
|---|---|---|---|
| Yes (**95.71%**) | Select from candidates through full analysis | 90.84 | 71.64 |
| | Reject and self-correct through full analysis | 4.87 | 1.73 |
| No (**4.29%**) | Select from candidates through superficial analysis | 3.93 | 2.28 |
| | Off-space prediction that bypasses candidates | 0.36 | 0.04 |
| Total (100%) | – | 100.00 | 75.69 |

As shown in Table 10, 95.71% of test cases involve full candidate-grounded analysis. This indicates that Retro-Expert usually examines the provided reactant candidates first and then makes a decision based on this analysis (i.e., selecting the correct candidates or rejecting and self-correcting incorrect ones). Meanwhile, hallucination and candidate bypass still occur in a small number of cases, accounting for 4.29% of the test cases.

Moreover, most correct predictions are obtained through candidate-grounded reasoning. This suggests that the policy induced by KGPO encourages the model to makes effective use of candidates while retaining the ability to recover from

incorrect candidates.

**Rejection & Self-Correction Behavior Analysis.** We further analyze whether KGPO encourages the LLM to perform critical behaviors (i.e., rejection and self-correction). Specifically, we compare the rejection and self-correction rates of GRPO and KGPO in the reasoning process on the test set.

*Table 11.* Analysis of candidate rejection and self-correction behaviors under different optimization settings.

| Setting | Reactant Candidate | Rejection Rate (%) | Overall Rejection Rate (%) | Self-Correction Rate (%) |
|---------|-------------------|-------------------|---------------------------|--------------------------|
| GRPO | Correct | 0.02 | 0.04 | 0.03 |
|  | Wrong | 0.22 |  |  |
| KGPO | Correct | 1.07 | 4.87 | 1.73 |
|  | Wrong | 39.12 |  |  |

As shown in Table 11, KGPO exhibits a higher rejection rate for incorrect candidates than GRPO, while maintaining a high selection rate for correct ones. This indicates that Retro-Expert can better discriminate candidate quality, enabling it to critically reject chemically implausible candidates while mostly preserving correct candidates.

The prompt used for training is as follows:

---

**System:**
You are a helpful AI Assistant that provides well-reasoned and detailed responses. You first think about the reasoning process as an internal monologue and then provide the user with the answer. Respond in the following format: `<think>\n...\n</think>\n<answer>\n...\n</answer>`

..............................................................................................................................

**Assistant:**
You are an experienced chemist analyzing chemical retrosynthesis.
Given the standard SMILES representation of the product is: {Product Standard SMILES},
Its automapping version is: {Product Mapped SMILES},
The IUPAC name of the product is: {IUPAC Name}.

The possible reaction type for synthesizing this product is: {Reaction Type}.
The possible reaction center for synthesizing this product is: {Reaction Center}.

**Reactants Candidate Set:**
(1) {Top-1 Reactants Candidate}
(2) {Top-2 Reactants Candidate}
(3) {Top-3 Reactants Candidate}
(4) {Top-4 Reactants Candidate}

Please reasonably infer the most likely reactants for synthesizing the product molecule from the Reactants Candidate Set. Please note that `<answer>\n...\n</answer>` should only contain the reactants.

---

1. The prompt for our Retro-Expert training.

## B. Details of Reasoning Process Evaluation

In this section, we describe the detailed setup of reasoning process evaluation. We evaluate the generated reasoning from two complementary aspects, (1) reasoning reliability and (2) explanation quality. Reasoning reliability examines whether the reasoning process faithfully supports the final prediction, including human-annotated reasoning correctness and decision consistency analysis. Explanation quality evaluates whether the generated explanation is chemically credible and understandable to experts, based on automatic and human scoring procedures.

### B.1. Reasoning Reliability Evaluation

We evaluate reasoning reliability through two analyses, reasoning correctness and decision consistency. We first provide the human annotation protocol for reasoning correctness, including the definition of correct reasoning, inter-annotator

agreement, and high-agreement case analysis. We then present the decision consistency analysis between intermediate decisions correctness and final predictions correctness.

### B.1.1. REASONING CORRECTNESS

We conduct a human evaluation to examine whether the generated reasoning is chemically sound and faithfully supports the final prediction. We randomly select 500 samples and ask three chemistry PhDs in organic synthesis to rigorously evaluate the model's reasoning. Each annotator was given the reasoning process produced by our model together with the ground-truth reaction information. To facilitate inspection, we also supplied molecular visualizations of the product and candidate reactants. They were asked to evaluate the reasoning process from multiple dimensions and provide a brief written justification for each dimension judgment. For example, "Incorrect. The reacting functional group in the product is an aryl bromide, not an alcohol."

We annotated each case along six dimensions:

- **Answer Correctness**: whether the final predicted reactants match the ground truth, labeled as correct or incorrect.
- **Product Analysis Correctness**: whether the analysis of the product structure is correct, partially flawed, or incorrect.
- **Candidate Decision Correctness**: whether the selected reaction type, reaction center, and reactants are correct, incorrect, or not explicitly stated.
- **Decision Rationale Correctness**: whether the rationale supporting each candidate decision is reasonable, partially flawed, or unreasonable.
- **Reasoning Faithfulness**: whether the reasoning process faithfully supports the final prediction, labeled as faithful, weakly faithful, or unfaithful.
- **Annotator Confidence**: whether the annotator has high or low confidence in the judgment.

A case is counted as overall correct reasoning only if all six dimensions are judged as correct, faithful, or high-confidence by at least two of the three annotators. This provides a stringent criterion for reasoning correctness.

*Table 12.* Human evaluation of reasoning correctness on 500 randomly selected samples.

| Answer | Reasoning Process | Coverage (%) |
| --- | --- | --- |
| Correct | Correct | 82.85 |
| Correct | Partially Correct | 17.15 |
| Wrong | Correct for a Plausible Alternative | 33.60 |
| Wrong | Partially Correct & Incorrect | 66.40 |

As shown in Table 12, correct predictions are mostly accompanied by chemically sound reasoning, whereas incorrect predictions more often contain flawed reasoning. This indicates that the generated reasoning is strongly aligned with answer correctness, rather than being independent post-hoc rationalization.

To further examine annotation reliability, we analyze cases where the experts agreed with each other, following the evaluation practice of GPQA. We report inter-annotator agreement across different annotation dimensions in Table 13, and further analyze the 3/3 agreed cases on the reasoning faithfulness dimension in Table 14.

*Table 13.* Inter-annotator agreement for reasoning correctness evaluation among three chemistry PhD annotators.

| Annotation Dimension | 3/3 Agreement (%) | 2/3 Agreement (%) |
| --- | --- | --- |
| Product Analysis Correctness | 80 | 99 |
| Candidate Decision Correctness (type, center, reactants) | 77, 85, 79 | 100, 100, 100 |
| Decision Rationale Correctness (type, center, reactants) | 73, 78, 71 | 98, 98, 96 |
| Reasoning Faithfulness | 83 | 98 |
| Overall Correct Reasoning | 59 | 63 |

As shown in Table 13, the annotations show high majority agreement across almost all dimensions. In particular, reasoning faithfulness achieves 83% unanimous agreement and 98% majority agreement, suggesting that human experts can consistently judge whether the reasoning faithfully supports the prediction. Furthermore, Table 14 shows that most high-agreement cases correspond to faithful & largely sound reasoning, including both correct answers and chemically plausible alternatives. This

*Table 14.* Fine-grained analysis of 3/3 agreed cases on reasoning faithfulness.

| Category | Percentage (%) |
|---|---|
| Fully sound reasoning with an answer matching the ground truth | 57 |
| Fully sound reasoning with a chemically plausible alternative answer | 6 |
| Largely sound but partially flawed reasoning with a correct answer | 16 |
| Weakly supported reasoning with a correct answer | 2 |
| Both incorrect answer and incorrect reasoning | 19 |

analysis provides a more transparent view of human evaluation than invoking expert judgment alone. It shows not only whether the reasoning is judged correct, but also where it remains faithful and where failures occur.

### B.1.2. DECISION CONSISTENCY

We also evaluate whether the intermediate decisions in the reasoning path are consistent with the final answer. Specifically, we examined the correctness of intermediate decisions (i.e. reaction type, center, reactants) in the model's reasoning on the test set.

*Table 15.* Decision consistency between the correctness of intermediate decisions and the correctness of final predictions in the reasoning on the USPTO-50K test set.

| Type | Center | Reactants | Final Answer | Coverage (%) |
|---|---|---|---|---|
| True | True | True | True | 64.06 |
| True | False | True | True | 2.84 |
| False | True | True | True | 8.79 |
| True | True | False | False | 0.38 |
| True | False | False | False | 4.57 |
| False | True | False | False | 2.01 |
| False | False | False | False | 17.35 |

As shown in Table 15, correct final answers are typically grounded in correct intermediate decisions, whereas incorrect answers are more frequently associated with erroneous decisions. This alignment indicates that the final prediction is tightly coupled with the structured reasoning path, suggesting that Retro-Expert reaches its conclusions through chemically valid reasoning.

### B.2. Explanation Quality Evaluation

In addition to reasoning reliability, we evaluate the quality of the natural language explanations in the reasoning process. This section details the prompts corresponding to the evaluation metrics, along with the automatic and human scoring methodologies.

### B.2.1. DETAILS OF EVALUATION METRICS

We provide all the prompts for evaluation.

---

**Mechanism Accuracy** ($\mathbb{MA}$)

**System:**

You are an expert in chemical retrosynthetic analysis, responsible for evaluating the quality of reasoning processes generated by retrosynthetic models.

#INSTRUCTIONS:

Based on the following the **Generated Reasoning Process**, conduct a professional assessment across the follow dimensions:

Mechanism Accuracy: Assess whether the reaction mechanism described in the reasoning aligns with established chemical principles.
[Score 1]: Incorrect mechanism.
[Score 2]: Partially correct but with significant deviations.
[Score 3]: Partially correct with minor deviations.
[Score 4]: Largely consistent.
[Score 5]: Fully consistent.

Evaluations must be grounded in given **Authentic Chemical Synthesis Information** and the results must be strictly formatted according to the specified Output Format.

[Output Format Example]
```
Mechanism Accuracy:2
```

.................................................................................................................

**Assistant:**

Please evaluate the mechanism accuracy of the **Generated Reasoning Process**, based on the given **Authentic Chemical Synthesis Information**.

**{Generated Reasoning Process}**

**Authentic Chemical Synthesis Information**:

The product's standard SMILES representation is {Product Standard SMILES}; Its automapping version is {Product Mapped SMILES}; The IUPAC name of the product is {IUPAC Name}.
The reaction type for synthesizing this product is: {The Golden Reaction Type}.
The reaction center of the product molecule is: {The Golden Reaction Center}.
The reactants for synthesizing the product molecule are: {The Golden Reactants}.

6. The prompt for Mechanism Accuracy ($\mathbb{MA}$).

---

**Factual Correctness** ($\mathbb{FC}$)

**System:**

You are an expert in chemical retrosynthetic analysis, responsible for evaluating the quality of reasoning processes generated by retrosynthetic models.

#INSTRUCTIONS:

Based on the following the **Generated Reasoning Process**, conduct a professional assessment across the follow dimensions:

Factual Correctness: Evaluate the presence of factual errors concerning chemical principles within the reasoning description.
[Score 1]: Major factual errors.
[Score 2]: Several errors.
[Score 3]: minor error(s).
[Score 4]: Essentially free of errors.
[Score 5]: Completely accurate.

Evaluations must be grounded in given **Authentic Chemical Synthesis Information** and the results must be strictly formatted according to the specified Output Format.

[Output Format Example]
```
Factual Correctness:2
```

...................................................................................................................

**Assistant:**

Please evaluate the factual correctness of the **Generated Reasoning Process**, based on the given **Authentic Chemical Synthesis Information**.

**{Generated Reasoning Process}**

**Authentic Chemical Synthesis Information**:

The product's standard SMILES representation is {Product Standard SMILES}; Its automapping version is {Product Mapped SMILES}; The IUPAC name of the product is {IUPAC Name}.
The reaction type for synthesizing this product is: {The Golden Reaction Type}.
The reaction center of the product molecule is: {The Golden Reaction Center}.
The reactants for synthesizing the product molecule are: {The Golden Reactants}.

7. The prompt for Factual Correctness ($\mathbb{FC}$).

**Logical Consistency** ($\mathbb{LC}$)

**System:**

You are an expert in chemical retrosynthetic analysis, responsible for evaluating the quality of reasoning processes generated by retrosynthetic models.

#INSTRUCTIONS:

Based on the following the **Generated Reasoning Process**, conduct a professional assessment across the follow dimensions:

Logical Consistency: Determine if conclusions follow logically from premises, analyses, or assumptions without contradictory leaps.

[Score 1]: Conclusion contradicts premises/analysis.

[Score 2]: Partially consistent but with significant logical leaps.

[Score 3]: Partially consistent with only minor leaps.

[Score 4]: Largely fluent and natural.

[Score 5]: Completely fluent and natural.

Evaluations must be grounded in given **Authentic Chemical Synthesis Information** and the results must be strictly formatted according to the specified Output Format.

[Output Format Example]
```
Logical Consistency:2
```

.........................................................................................................................

**Assistant:**

Please evaluate the logical consistency of the **Generated Reasoning Process**, based on the given **Authentic Chemical Synthesis Information**.

**{Generated Reasoning Process}**

**Authentic Chemical Synthesis Information**:

The product's standard SMILES representation is {Product Standard SMILES}; Its automapping version is {Product Mapped SMILES}; The IUPAC name of the product is {IUPAC Name}.

The reaction type for synthesizing this product is: {The Golden Reaction Type}.

The reaction center of the product molecule is: {The Golden Reaction Center}.

The reactants for synthesizing the product molecule are: {The Golden Reactants}.

8. The prompt for Logical Consistency ($\mathbb{LC}$).

### B.2.2. DETAILS OF HUMAN STUDY

Based on the USPTO-50K test set, we provide the model's interpretable reasoning process along with corresponding ground truth including: reactant-product SMILES, reaction type, and reaction centers. Evaluators were first presented with ground truth information for each sample, then asked to score the interpretable reasoning process. Specifically, they rate the reasoning in terms of (1) Mechanism Accuracy, (2) Fatual Correctness, and (3) Logical Consistency. Each criterion is scored on a scale from 1 to 5, with 5 indicating the highest quality. The final score for a sample is calculated as the average of the scores assigned by all evaluators. The evaluation interface layout is shown in Figure 5.

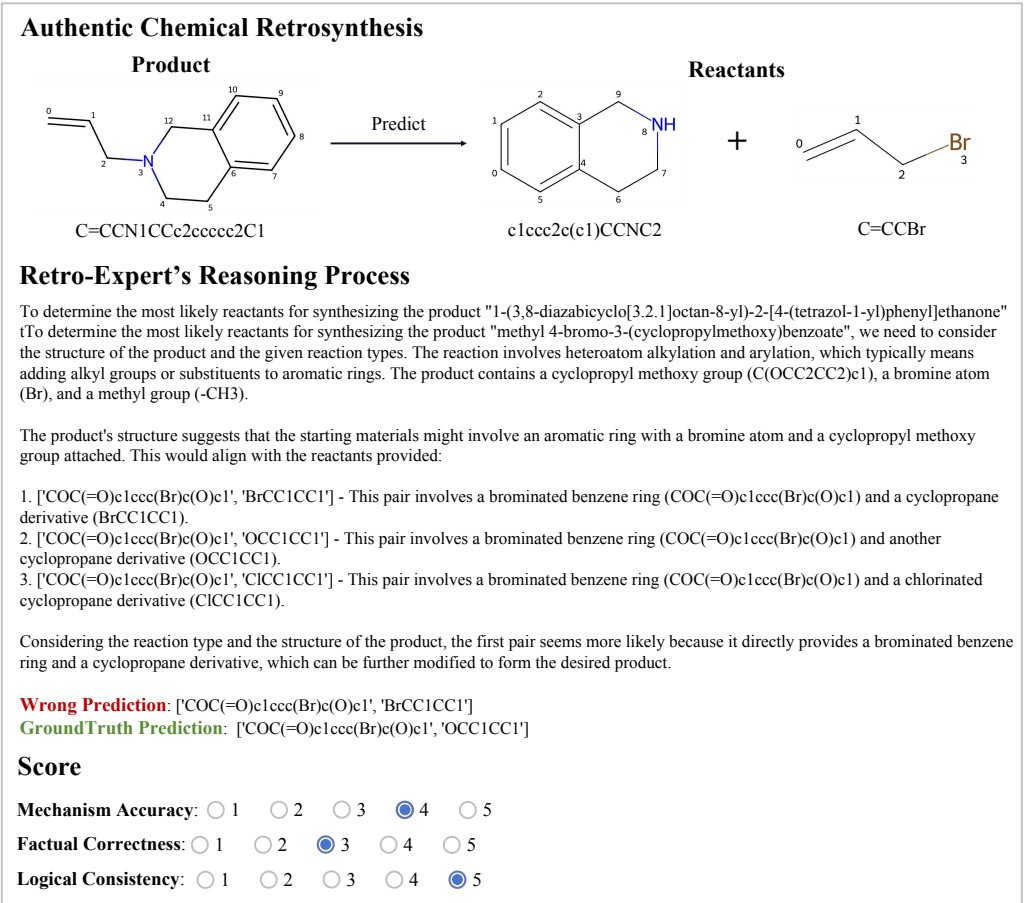

*Figure 5.* Generated reasoning process rating for human evaluation — assessing the mechanism accuracy, fatual correctness, logical consistency of the generated reasoning process on a scale of 1 to 5, with higher scores indicating superior reasoning quality.

## C. Details of Multi-Step Retrosynthesis

In this section, we describe how Retro-Expert is adapted to multi-step retrosynthesis. We provide details on the search algorithm, the route-level decision space, and representative examples showing that the model performs step-aware reasoning beyond simple route reranking.

### C.1. Route-Level Decision Space for Multi-Step Planning

In the multi-step setting, Retro-Expert does not invoke the LLM at every single-step node expansion, as in MCTS routing. Instead, we decouple route search from reasoning. A multi-step specialized model (DMS-Model) first generates Top-K complete multi-step routes via beam search. These complete routes are then jointly provided to the LLM in a single query as a route-level decision space. Each candidate route contains the target product, step-wise predicted intermediates and final predictions. Specifically, each route starts from the target product and recursively decomposes each predicted intermediate

into its corresponding reactants. A candidate route is formatted as:

$$R_i = \{P_i, s_i^1, s_i^2, \ldots, s_i^{L_i}\},$$

where $P_i$ is the target product, and each step $s_i^j$ is represented as:

$$s_i^j : M_i^j \rightarrow \{R_{i,1}^j, R_{i,2}^j, \ldots\}.$$

Here, $M_i^j$ denotes the molecule to be decomposed at step $j$, while $\{R_{i,1}^j, R_{i,2}^j, \ldots\}$ denotes the predicted reactants for that intermediate.

For example, a candidate route can be represented as:

| | |
|---|---|
| Product: | `CC(C)(C=C(F)Cc1ccc(F)c(Oc2ccccc2)c1)c1ccc(Cl)cc1` |
| Step-1: | `CC(C)(C=C(F)Cc1ccc(F)c(Oc2ccccc2)c1)c1ccc(Cl)cc1` |
| | `>> OB(O)c1ccc(F)c(Oc2ccccc2)c1 and CC(C)(C=C(F)CBr)c1ccc(Cl)cc1` |
| Step-2: | `OB(O)c1ccc(F)c(Oc2ccccc2)c1` |
| | `>> Fc1ccc(Br)cc1Oc1ccccc1 and COB(OC)OC` |
| | `CC(C)(C=C(F)CBr)c1ccc(Cl)cc1` |
| | `>> BrBr and CC(C)(C=C(F)CO)c1ccc(Cl)cc1.` |
| Reactants: | `Fc1ccc(Br)cc1Oc1ccccc1 and COB(OC)OC and` |
| | `BrBr and CC(C)(C=C(F)CO)c1ccc(Cl)cc1` |

Given this step-wise route decision space, the LLM then performs a single round of reasoning (e.g. analyze, select, compare) over all candidate routes. Specifically, the LLM analyzes individual steps within each candidate route and compares routes globally, indicating both step and route-level reasoning. For example, "I rule out Route-1 because its Step-2 product lacks the aldehyde required for the subsequent oxidation." This design assigns efficient route search to the specialized model and reserves high-value chemical reasoning for the LLM, yielding improved accuracy at practical cost.

### C.2. Beyond Reranking by Step-Aware Reasoning

Retro-Expert performs step-aware route reasoning over the candidates. Specifically, **our model reasons at two levels**. At the step level, it examines whether the transformation in each step is chemically plausible and whether the predicted intermediate is consistent with the proposed reactants and reaction logic. At the route level, it compares different routes based on step-level analysis. For example, "I rule out Route-A because its Step-2 product lacks the aldehyde required for the subsequent oxidation."

Therefore, Retro-Expert can exploit partially correct local decisions across different routes and combine them into a better final answer (e.g., using one Step-2 branch from Route-A and another Step-2 branch from Route-B to form the Step-2 in the new route). It can also generate a new route conditioned on the candidate routes when it finds none of them satisfactory. This behavior goes beyond conventional reranking, which typically scores each complete route as a whole.

We provide a representative example on the synthesis of 2-phenylethyl benzoate (`C1=CC=C(C=C1)CCOC(=O)C2=CC=CC=C2`). In this case, Retro-Expert does not simply select one complete route. Instead, it combines locally plausible step-wise decompositions from different routes. It keeps the Step-2 branch from Route-A to produce 2-phenylethanol (`c1ccc(cc1)CCO`) from ethylene oxide (`C1CO1`) and phenylmagnesium bromide (`Br[Mg]c1ccccc1`). Meanwhile, it adopts the Step-2 branch from Route-B to obtain benzoic acid (`c1ccccc1C(=O)O`) from toluene (`Cc1ccccc1`) via potassium permanganate oxidation. Our model rejects Route-A's benzoic-acid branch as more demanding and complex, stating: "This step demands harsh reaction conditions due to strong acid catalysis... It appears more complex than Route-B for the preparation of benzoic acid." Therefore, the final route is obtained through both step-level and route-level reasoning, rather than by simply reranking complete candidate routes.

## D. Details of Web Lab Experiments

To validate Retro-Expert's practical utility in real-world chemical discovery, we systematically describe the case selection criteria and experimentally verify two predicted synthetic routes. Sci-Finder serves as the gold standard for determining route novelty.

### D.1. Case Selection Criteria

For the wet-lab study, we first collected the Top-K predictions from Retro-Expert on the test set, and then selected experimental cases based on three criteria:

- **Novelty.** We prioritized cases with unreported synthetic routes.

- **Experimental feasibility.** We retained only cases with accessible starting materials, manageable time and cost.

- **Diversity.** Among feasible candidates, we selected examples that reflect different practical application scenarios.

The following proposed two cases illustrate two key application scenarios: discovering a new route and designing an alternative route. Both routes are the model's Top-1 predictions.

### D.2. Synthesis of 3-(2-ethoxyphenyl)thiophene

Retro-Expert predicts the target molecule could be synthesized via Suzuki-Miyaura coupling between 1-bromo-2-ethoxybenzene and thiophen-3-ylboronic acid. The reaction scheme is visualized in Figure 6. Furthermore, the occurrence of the reaction and the correct structure of the synthesized product are confirmed by both the $^1$H NMR and $^{13}$C NMR spectra, as shown in Figure 7.

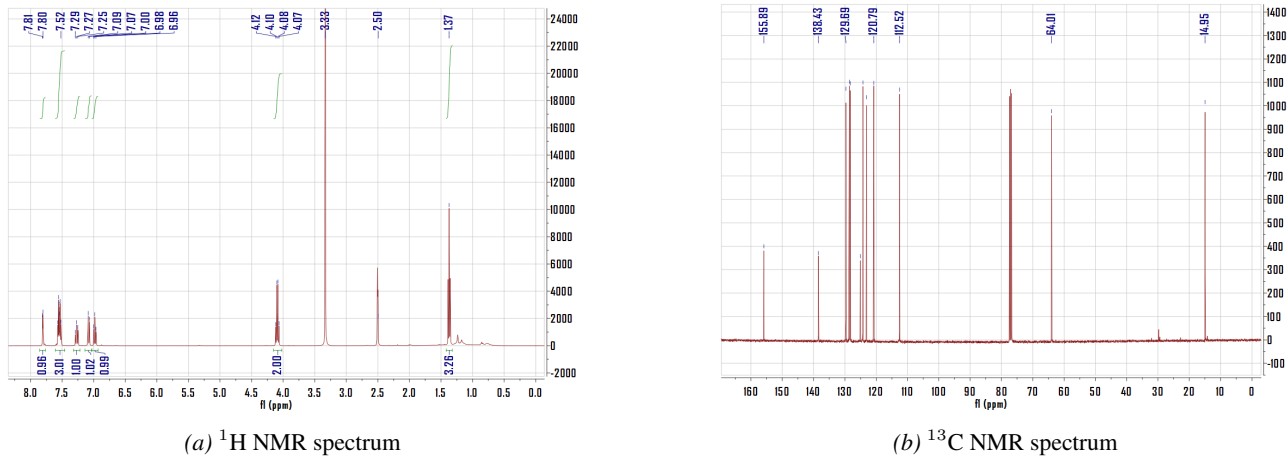

*Figure 6.* The synthesis path of 3-(2-ethoxyphenyl)thiophene.

*(a)* $^1$H NMR spectrum

*(b)* $^{13}$C NMR spectrum

*Figure 7.* NMR characterization of the synthesized product 3-(2-ethoxyphenyl)thiophene. Both $^1$H and $^{13}$C NMR spectra support the successful formation and structural integrity of the compound.

The detailed experimental procedure: 1-bromo-2-ethoxybenzene (0.20 g, 1.0 mmol, M1), 3-thiopheneboronic acid (0.15 g, 1.2 mmol, M2), and tetrakis (triphenylphosphine)palladium(0) (0.12 g, 0.1 mmol) were added to a 25 mL three-necked flask. The flask was evacuated and backfilled with nitrogen three times to ensure an oxygen-free atmosphere. An aqueous solution of $K_2CO_3$ (1 M) was prepared and set aside. Subsequently, 8 mL of tetrahydrofuran (THF) was added to the flask, followed

by the addition of 2 mL prepared $K_2CO_3$ solution, ensuring the reaction system remained under a continuous nitrogen purge throughout the addition. The reaction mixture was heated to reflux at 92 °C under a nitrogen atmosphere. Reaction progress was monitored by thin-layer chromatography (TLC). After 40 hours, the reaction was deemed complete. The excess solvent was removed under reduced pressure using a rotary evaporator. The crude residue was diluted with 50 mL of water and extracted with 50 mL of ethyl acetate. The layers were separated, and the extraction process with ethyl acetate was repeated three times. The combined organic extracts were dried over anhydrous $Na_2SO_4$. After filtration, the solvent was removed under reduced pressure to afford the crude product as a dark brown liquid. The crude material was purified directly by column chromatography using petroleum ether as the eluent, yielding the desired product as a pale yellow liquid (0.16 g, 79.3%).

### D.3. Synthesis of 1-(4-ethoxyphenyl)ethanone

Retro-Expert identified a novel route via Jones oxidation. The synthesis proceeds from 1-(4-ethoxyphenyl)ethanol under acidic conditions using chromium trioxide. A visual representation of the reaction process is provided in Figure 8. Furthermore, the successful oxidation of the secondary alcohol to the corresponding ketone and the formation of the correct product structure were confirmed by $^1$H NMR spectroscopy.

$^1$H NMR spectrum of the synthesized 1-(4-ethoxyphenyl)ethanone, confirming the successful oxidation of the secondary alcohol to a ketone. The observed chemical shifts are consistent with the expected structure.

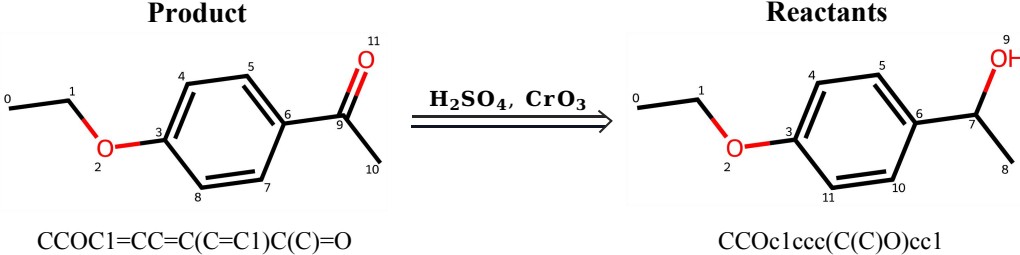

*Figure 8.* The synthesis path of 1-(4-ethoxyphenyl)ethanone.

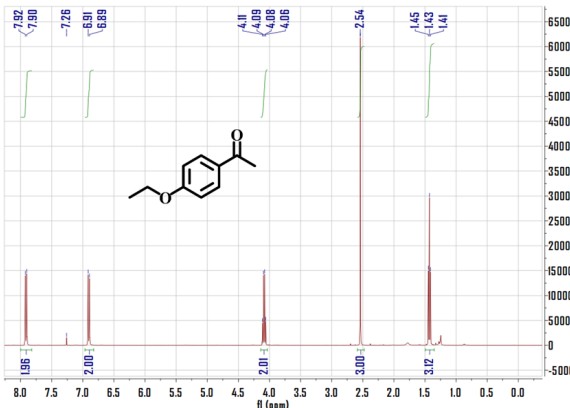

*Figure 9.* $^1$H NMR spectrum of the synthesized 1-(4-ethoxyphenyl)ethanone, confirming the successful oxidation of the secondary alcohol to a ketone. The observed chemical shifts are consistent with the expected structure.

The detailed experimental procedure: Chromium trioxide (1.80 g, 0.018 mol) was dissolved in 8.2 mL of deionized water in a 20 mL sample vial. Concentrated sulfuric acid (1.8 mL) was then added dropwise with continuous stirring while maintaining the reaction temperature at 0 °C throughout. The resulting orange-red solution constituted the *Jones reagent* (1.8 M). Separately, 1-(4-ethoxyphenyl)ethanol (0.22 g, 0.013 mol, J1) was dissolved in 10 mL of cold acetone (0 °C) in a 25 mL round-bottom flask. While keeping the temperature at 0 °C, 2 mL of the freshly prepared Jones reagent was added dropwise. The mixture was stirred at 0 °C for 3 hours. Subsequently, 10 mL of ice-cold saturated $NaHSO_3$ solution was added to quench the reaction, and the pH was adjusted to 13 using ice-cold aqueous KOH. The resulting mixture was extracted with chloroform (3 × 50 mL) and washed with deionized water. The combined organic layers were dried over

anhydrous $Na_2SO_4$ and concentrated under reduced pressure at room temperature, yielding 125.5 mg of a pale yellow oily product with a yield of 58.82%.

## E. Additional Ablation Study

This section presents additional ablation studies.

**KGPO Gains from Both Chem-RM and Rectified Optimization.** To isolate the contribution of KGPO, we compare GRPO and KGPO under matched reward settings. KGPO contributes in two ways: (1) Reward Modeling (in Eq. (3)): standard GRPO uses only the final-answer reward $r_{base}$, which provides sparse supervision does not directly evaluate intermediate chemical decisions. In our KGPO setting, we introduce Chem-RM, a reward model that utilizes the chemical toolkit (e.g., RDKit) to provide dense and accurate reward signals grounded in intrinsic deterministic chemical knowledge. It avoids the known collapse issues of LLM-as-a-judge evaluation pipelines. (2) Algorithm Optimization (in Eq. (6)): standard GRPO computes advantages via linear group normalization, which can underweight low-frequency yet high-value correction behaviors. By contrast, KGPO increases the relative advantage of trajectories that correct erroneous candidates, encouraging the LLM to evaluate candidates critically rather than follow them blindly.

*Table 16.* Ablation study of GRPO and KGPO under different reward models and optimization strategies.

| Setting | Optimization Strategy | Reward Model | Top-1 Acc. (%) |
|---------|----------------------|--------------|----------------|
| GRPO | Standard optimization | $r_{base}$ | 66.8 |
| | Standard optimization | $r_{base} + r_{react}$ | 69.9 |
| | Standard optimization | Chem-RM | 73.1 |
| KRPO | Rectified optimization | $r_{base}$ | 66.8 |
| | Rectified optimization | $r_{base} + r_{react}$ | 72.3 |
| | Rectified optimization | Chem-RM | 75.7 |

As shown in Table 16, the full-system gain of KGPO over standard GRPO is 8.9 points, improving Top-1 accuracy from 66.8% to 75.7%. The 6.3-point gain reflects the contribution of dense chemically grounded reward modeling, while the additional 2.6-point gain comes from the rectified update of KGPO. These results show that KGPO benefits from the synergy between reward modeling and rectified optimization.

**Effective Reasoning Hinges on the Chemical Decision Space.** To demonstrate the necessity and effectiveness of constructing chemical decision space, we train LLMs using KGPO under different decision space. As shown in Table 17, when only four candidate reactants are provided, the model's prediction accuracy is extremely low (29.1%). When gradually add "type", "center" into the decision space, the performance gradually increases to 75.7%. This indicates that a structured, multi-dimensional decision space is essential to properly ground the LLM's logical reasoning and achieve high accuracy.

*Table 17.* Performance comparison under different decision space based on KGPO training policy.

| Decision Space | | | Reactants Top-1 Accuracy (%) |
|------|--------|----------|------------------------------|
| Type | Center | Reactant | |
| | | ✓ | 29.1 |
| ✓ | | ✓ | 34.2 |
| | ✓ | ✓ | 57.8 |
| ✓ | ✓ | ✓ | **75.7** |

**Varying N Candidate.** We analyze the impact of using different N values for candidate results during inference on LLM reasoning performance in Table 18. As N increases, the Top-1 reactant prediction accuracy gradually improves. When five or more candidates are provided, the model's performance stabilizes at 76.1%. This indicates that the model can effectively perform critical reasoning based on the candidate set and generate the optimal answer. To ensure a fair comparison while maintaining competitive performance, we use the Top-4 candidate for reactant-related decision space, as chemical LLMs such as ChemLLM rely on four options for selection.

**Reward hacking.** Due to the probabilistic bias in the Varying-N candidate results provided by small models, i.e., the correct answer appears in the Top-1 position with significantly higher probability than in other positions. When using result correctness as the reward, the LLM will directly select the first candidate to continuously maximize its reward. To validate

*Table 18.* The effect of the N value in small models' N candidates on large model performance. $\Delta$ denotes the improvement relative to the Top-1 accuracy of small models' reactant prediction (67.2%).

| N Candidates | 1 | 3 | 4 | 5 | 8 |
|---|---|---|---|---|---|
| Reactant Top-1 | 70.1 | 73.5 | 75.7 | 76.1 | 76.1 |
| $\Delta$ | **+2.9** | **+6.3** | **+8.5** | **+8.9** | **+8.9** |

the effectiveness of our candidate position shuffling strategy, we visualized the positional distribution of correct answers in the small model's N candidates and the distribution of positions selected by the LLM. As shown in Figure 10, using the original N candidates from small models results in severe reward hacking, whereas our position shuffling strategy effectively mitigates this issue.

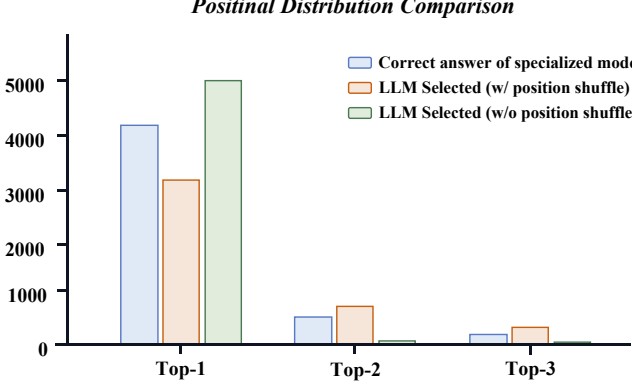

*Figure 10.* Comparison of positional distributions between the correct answers of small models and the selected positions of LLMs.

# F. Results Analysis

This section showcases visualized examples illustrating Retro-Expert's reasoning process, highlights both successful and failed cases. For readability, we present condensed reasoning traces that provide a brief view of the reactant-level decision-making process.

### F.1. Successful Reasoning (Selection Scene)

We demonstrate the reasoning processes generated by our Retro-Expert across diverse reaction types (in Figure 11-13). This indicates Retro-Expert's fundamental competency in executing chemically valid reasoning and making accurate selections within a high-quality chemical action space.

Specifically, our Retro-Expert is capable of performing effective critical reasoning over all candidate options rather than merely selecting the top-1 prediction. During the decision-making process, the model actively incorporates key reaction-related information—such as reaction type and center—to infer plausible mechanisms and eliminate implausible candidates. Notably, Retro-Expert demonstrates strong generalization even for reaction types with limited representation in the training data, and maintains high accuracy across molecules with varying structural complexity. This highlights its ability to perform nuanced, context-aware retrosynthetic reasoning beyond surface-level pattern recognition.

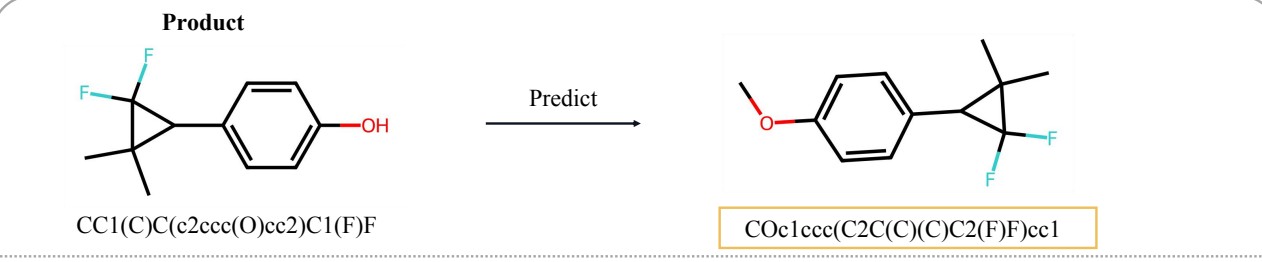

**Product**

CC1(C)C(c2ccc(O)cc2)C1(F)F

Predict →

COc1ccc(C2C(C)(C)C2(F)F)cc1

Reactant

🗣 **Retro-Expert Reasoning Process Snapshot:**

<think>

**Reaction Type**

To determine the most likely reactants for synthesizing the product molecule, we need to consider the given information and the reaction type provided. The product has a deprotection reaction type, which means it involves the removal of a protective group. From the IUPAC name, we can deduce that the protective group on phenol (OH) is being removed to form the desired phenol structure.

**Reaction Center**

The product's SMILES is "CC1(C)C(c2ccc(O)cc2)C1(F)F", and its automapping version suggests a complex structure with fluorine and methyl groups. This implies that the starting material must have these groups attached to the phenol ring. Looking at the Reactants Candidate Set:

1. ['COc1ccc(C2C(C)(C)C2(F)F)cc1'] - This represents a compound with a cyclopropane derivative, where the cyclopropane carbon atoms are attached to two fluorine atoms.
2. ['CC1(C)C(c2ccc(OCc3ccccc3)cc2)C1(F)F'] - This is the product we are trying to synthesize, so it is not a suitable starting material.
3. ['COCOc1ccc(C2C(C)(C)C2(F)F)cc1'] - This compound also contains a cyclopropane derivative but does not have any phenol or OH groups attached, making it less likely to be a direct precursor.

Considering the deprotection reaction type, the most plausible starting material would be one that has a protective group (such as a tert-butyl group or a benzyl group) on the phenol ring. However, since the product already contains fluorine and methyl groups, it seems more likely that the starting material would have these groups attached to a different structure that could be deprotected to form the product.

**Reactant**

Given the structure of the starting material, the first candidate ['COc1ccc(C2C(C)(C)C2(F)F)cc1'] appears promising. This structure contains a cyclopropane derivative, which might undergo deprotection to form the desired phenol structure. The second candidate ['CC1(C)C(c2ccc(OCc3ccccc3)cc2)C1(F)F'] is the product itself and cannot be used as a starting material. The third candidate ['COCOc1ccc(C2C(C)(C)C2(F)F)cc1'] lacks phenol functionality and thus is unlikely to be a viable starting material.

Thus, the most likely reactant for synthesizing the product is the first candidate:
</think>

*Figure 11.* Visualization of reasoning process (1/3) for oxidation reaction.

**Product**

CCc1c(NC2CC2)c([N+](=O)[O-])cc(F)c1N1CCN(C)CC1

Predict →

CCc1c(F)c([N+](=O)[O-])cc(F)c1N1CCN(C)CC1

+

NC1CC1

Reactant

🧠 **Retro-Expert Reasoning Process Snapshot:**

Reaction Type

<think>

To determine the most likely reactants for synthesizing the product, we need to consider the given information and the reaction type specified. The product is a complex aromatic amine derivative, and the reaction type is "heteroatom alkylation and arylation". This implies that we are likely dealing with a substitution or addition reaction involving a heteroatom (likely nitrogen or oxygen).

Given the IUPAC name of the product "N-cyclopropyl-2-ethyl-4-fluoro-3-(4-methylpiperazin-1-yl)-6-nitroaniline", we can deduce that the starting material needs to be an aniline with multiple substituents. The substituents include a cyclopropylamino group, ethyl, fluoro, methylpiperazine, and nitro groups.

Looking at the reactants candidate set:

1. ['CN1CCNCC1', 'CCc1c(F)c(F)cc([N+](=O)[O-])c1NC1CC1']

   - The first compound is N-methylpiperazine.

   - The second compound is a benzene ring with multiple substituents including fluorine, nitro, and a cyclopropylamino group.

2. ['CCc1c(F)c([N+](=O)[O-])cc(F)c1N1CCN(C)CC1', 'NC1CC1']

   - The first compound is a complex fluoro-nitro aromatic derivative bearing the methylpiperazine substituent.

   - The second compound is cyclopropylamine.

3. ['CCc1c(Br)c(F)cc([N+](=O)[O-])c1NC1CC1', 'CN1CCNCC1']

   - The first compound is a benzene ring with a bromine atom and multiple substituents.

   - The second compound is N-methylpiperazine.

Reaction Center                                              Reactant

Considering the complexity of the product and the need to form the aryl C–N bond at the reaction center, the second reactant set seems more promising because it includes a complex fluoro-nitro aromatic substrate and cyclopropylamine, which could form the desired product through heteroatom arylation.

Based on these considerations, the most likely reactants for synthesizing the product are:

</think>

*Figure 12.* Visualization of reasoning process (2/3) for deprotection reaction.

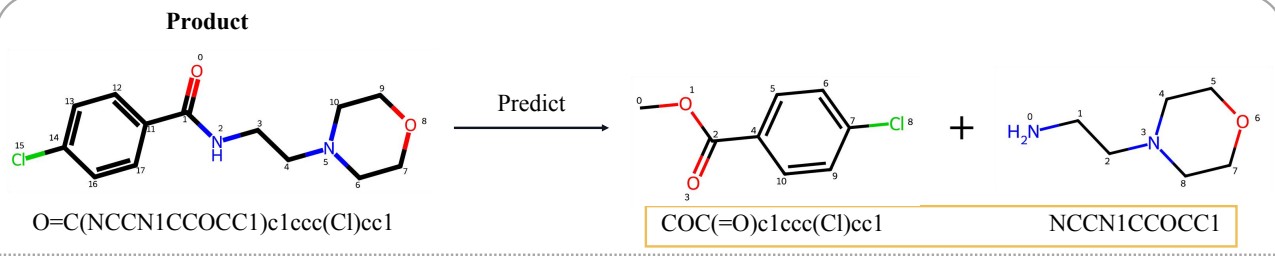

**Product**

O=C(NCCN1CCOCC1)c1ccc(Cl)cc1

Predict

COC(=O)c1ccc(Cl)cc1 + NCCN1CCOCC1

*Reactant*

🧠 **Retro-Expert Reasoning Process Snapshot:**

\<think\>

To determine the most likely reactants for synthesizing the product, we need to consider the functional groups present in the product's structure and the common synthetic routes used to form these groups. The given product is "4-chloro-N-(2-morpholin-4-ylethyl)benzamide". This compound contains a benzoyl amide group, a morpholine ring, and a chlorine atom.

From the SMILES and automapping versions, we can see that the amide C–N bond is part of the product, which suggests that we might be starting with an acyl donor and the amine component.

*Reaction Center*

Let's analyze the reactants:

1. ['O=C(O)c1ccc(Cl)cc1', 'NCCN1CCOCC1'] - This pair consists of 4-chlorobenzoic acid and a morpholine derivative.

2. ['NCCN1CCOCC1', 'O=C(Cl)c1ccc(Cl)cc1'] - This pair consists of the morpholine derivative and 4-chlorobenzoyl chloride.

3. ['NCCN1CCOCC1', 'COC(=O)c1ccc(Cl)cc1'] - This pair consists of the morpholine-containing amine and methyl 4-chlorobenzoate.

*Reaction Type*

Considering the reaction type is acylation and related processes, we can infer that the key to forming the benzamide group is to start with an acyl donor precursor. Let's evaluate each option:

1. In the first pair, we have 4-chlorobenzoic acid ('O=C(O)c1ccc(Cl)cc1') and a morpholine derivative ('NCCN1CCOCC1'). This could make sense if the carboxylic acid is activated, but it is not the most direct acyl donor among the candidates.

2. In the second pair, we have the morpholine derivative and 4-chlorobenzoyl chloride ('O=C(Cl)c1ccc(Cl)cc1'). While this could potentially work, it is not the candidate that matches the acyl donor structure implied by the automapping.

3. In the third pair, we have the morpholine derivative and a methyl ester acyl donor ('COC(=O)c1ccc(Cl)cc1'). Unlike the second pair, it is suitable because the amine can attack the ester carbonyl to form the amide bond, with methanol as the leaving group.

Given these considerations, the most plausible and straightforward route would involve starting with the methyl ester acyl donor and the morpholine derivative step by step, which aligns with the third pair of reactants.

\</think\>

*Reactant*

*Figure 13.* Visualization of reasoning process (3/3) for heteroatom alkylation and arylation reaction.

## F.2. Successful Reasoning (Generation Scene)

Furthermore, we showcase Retro-Expert's capacity for self-reflection and generation. Crucially, when confronted with suboptimal candidates from all specialized models, Retro-Expert engages in critical analysis and autonomously synthesizes a novel, chemically sound solution (in Figure 14-16).

This generative capability is observed in two distinct yet challenging scenarios. When the candidate set is empty (i.e., no viable options are proposed by the specialized models), Retro-Expert proactively initiates a reasoning process based on available information in the decision space (e.g., reaction type and center). Through this process, it constructs a plausible synthetic route grounded in chemical principles. When all candidate options are present but incorrect, Retro-Expert strategically rejects them and synthesizes an improved solution by critically analyzing the flaws in the proposed candidates and leveraging useful fragments or partial patterns they contain. In both settings, the model demonstrates a high level of autonomy and domain-aware creativity, producing chemically valid outputs even in the absence of reliable reactant candidate.

**Product**

Predict

+

N=C(N)c1ccc2[nH]c(C3([N+](=O)[O-
])C=C(c4nnn[nH]4)C=C(c4ccccc4)C3O)cc2c1

CC(=O)C1([N+](=O)[O-
])C=C(c2nnn[nH]2)C=C(c2ccccc2)C1O

N=C(N)c1ccc(NN)cc1

Reactant

🧠 **Retro-Expert Reasoning Process Snapshot:**

Reaction Type

<think>

To infer the most likely reactants for synthesizing the given product, we need to consider the key functional groups and heteroatoms present in the IUPAC name of the product. The IUPAC name "2-[6-hydroxy-1-nitro-5-phenyl-3-(2H-tetrazol-5-yl)cyclohexa-2,4-dien-1-yl]-1H-indole-5-carboximidamide" indicates several important groups:

- A hydroxy group (-OH) at position 6 of a cyclohexa-2,4-diene.
- A nitro group (-NO2) at position 1 of the same compound.
- A 2H-tetrazol-5-yl group attached to a phenyl ring.
- An indole moiety with a carboximidamide group at the 5-position.

Given these functional groups, let's identify potential starting materials and reaction steps:

1. **Substituted cyclohexadienone**: A substituted cyclohexadienone bearing a nitro, hydroxy, tetrazolyl, and phenyl group can be derived from a suitably functionalized acetophenone derivative through nitration and Michael-type addition reactions.

2. **Indole with carboximidamide**: The indole scaffold bearing a carboximidamide group can be derived from a substituted aniline precursor, such as a hydrazine-functionalized aniline, which enables subsequent cyclization to indole and installation of the amidine group.

Considering the above, the most likely reactants would be:

Reactant

- A substituted acetophenone derivative containing a nitro, phenyl, hydroxy, and tetrazolyl group
- A hydrazinyl-substituted benzamidine derivative , which can undergo condensation and cyclization to form the indole moiety.

These reactants can be combined through suitable reactions such as aldol condensation, nitration, and cyclization to form the desired product.
</think>

*Figure 14.* Visualization of the reasoning process (1/3) when no candidate is available.

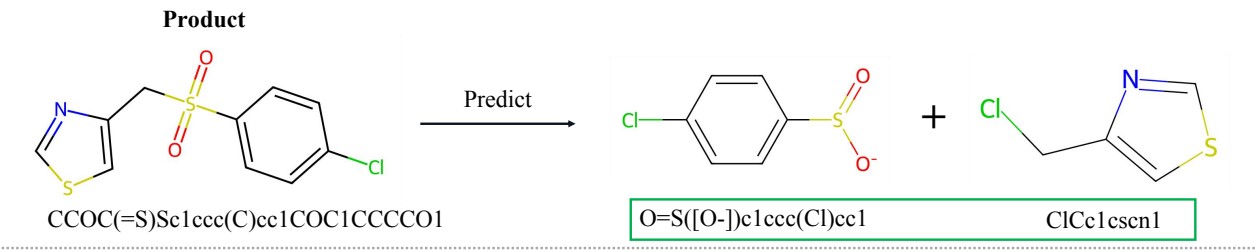

**Product**

CCOC(=S)Sc1ccc(C)cc1COC1CCCCO1

O=S([O-])c1ccc(Cl)cc1          ClCc1cscn1

Golden Reactant

### 🗣 Retro-Expert Reasoning Process Snapshot:

<think>

To infer the most likely reactants for synthesizing the given product, we need to consider the structure of the target compound, which is "4-[(4-chlorophenyl)sulfonylmethyl]-1,3-thiazole". The key components are:

- Sulfonylmethyl group (R-SO2CH3)
- 1,3-thiazole ring
- 4-Chlorophenyl substituent

The sulfonylmethyl group can be derived from a sulfonic acid or sulfonate, which can react with a halomethyl-substituted heterocycle to form a sulfone linkage via nucleophilic substitution. The 1,3-thiazole ring is likely introduced as a pre-functionalized unit bearing a reactive substituent such as a chloromethyl group.

Reaction Type

Given the reaction type "heteroatom alkylation and arylation", we need to identify a suitable starting material that can be modified to form both the sulfonylmethyl group and the thiazole ring.

Considering these points, One possible reactant could be a chloromethyl-substituted thiazole that reacts directly with a sulfonic acid or sulfonate derivative to form the sulfonylmethyl linkage.

Let's consider the reactants that might be used in such a synthesis:

Generated Reactant

1. A chloromethyl-substituted thiazole (e.g., $ClCH_2$-thiazole) acting as the alkylating agent.
2. A sulfonic acid or sulfonate derivative (e.g., para-chlorobenzenesulfonate) serving as the nucleophile.

One plausible candidate for the initial reaction step could be a chlorinated thiazole derivative that can be dechlorinated and then reacted with a sulfonyl group to form the desired sulfonylmethyl group and thiazole ring.

</think>

*Figure 15.* Visualization of the reasoning process (2/3) when no candidate is available.

**Product**

C[C@H](NC(=O)OC(C)(C)C)[C@@H](O)C
NS(=O)(=O)c1ccccn1

C[C@H](NC(=O)OC(C)(C)C)[C@H](O)CN

O=S(=O)(Cl)c1ccccn1

Reactant

🧠 Retro-Expert Reasoning Process Snapshot:

<think>

To infer the most likely reactants, we need to consider the structure of the target product and the given reactant candidates. The product has a tert-butyl group (-C(CH3)3), an amide (-CONH-) functional group, and a pyridine sulfonyl group (-SO2N(Pyridine)).

Looking at the reactant candidates:

1. ['O=S(=O)(Cl)c1ccccn1', 'C[C@H](NC(=O)OC(C)(C)C)[C@@H](O)CN']
   - The first component is a sulfonyl chloride (SO2Cl).
   - The second component matches the structure of the product's backbone.
2. ['O=S(=O)(O)c1ccccn1', 'C[C@H](NC(=O)OC(C)(C)C)[C@@H](O)CN']
   - The first component is a sulfonyl oxide (SO2O).
   - The second component matches the structure of the product's backbone.
3. ['O=S(=O)(Br)c1ccccn1', 'C[C@H](NC(=O)OC(C)(C)C)[C@@H](O)CN']
   - The first component is a sulfonyl bromide (SO2Br).
   - The second component matches the structure of the product's backbone.

Given the presence of the tert-butyl group, which typically involves a tertiary carbon bonded to three alkyl groups, and the need to form the amide bond, the most plausible scenario would involve an acylation reaction where the sulfonyl group reacts with the primary amine in the product.

Reaction Type          Reaction Center

In the context of the reaction types and centers provided, it seems the sulfonyl chloride (SO2Cl) or sulfonyl oxide (SO2O) might be more appropriate as they can more readily form the ester bond needed for the tert-butyl carbamate formation. The sulfonyl bromide (SO2Br) could also potentially react, but it might be less favorable due to its weaker acidity compared to the chloride and oxide forms.

Reject Options and Generation

While sulfonyl chloride remains a strong candidate for delivering the sulfonyl moiety, the backbone structure of the nucleophilic partner may require closer scrutiny to ensure stereochemical consistency. Given the importance of preserving chiral integrity, it would be reasonable to explore a variant of the backbone reactant that aligns precisely with the stereochemistry observed in the product. This suggests a slightly adjusted molecular involving this stereochemically consistent intermediate and a pyridyl sulfonyl chloride.

Generated Reactant

A more plausible disconnection involves reacting a primary amine-containing intermediate — which precisely preserves the required stereochemistry — with a pyridine-based sulfonyl chloride to form the final sulfonamide linkage under mild conditions.
</think>

*Figure 16.* Visualization of the reasoning process (3/3) when the provided candidate is incorrect.

## F.3. Failure Analysis

This section presents examples of model prediction failures. As shown in Figure 17, it reveals two scenarios encountered by our Retro-Expert: (1) Multiple chemically similar reactive sites within a molecule, (2) Multiple distinct transformation pathways under the same reaction type containing broad reaction. These limitations indicate the model's current inability to perform site-specific chemical discrimination. Future work will integrate domain-specific chemical heuristics and specialized physical evaluation models (energetic/kinetic).

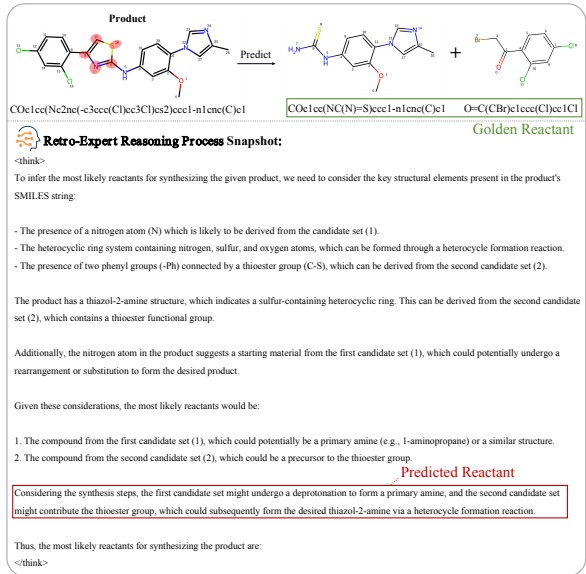

*(a)* Failure case for multiple reactive sites.

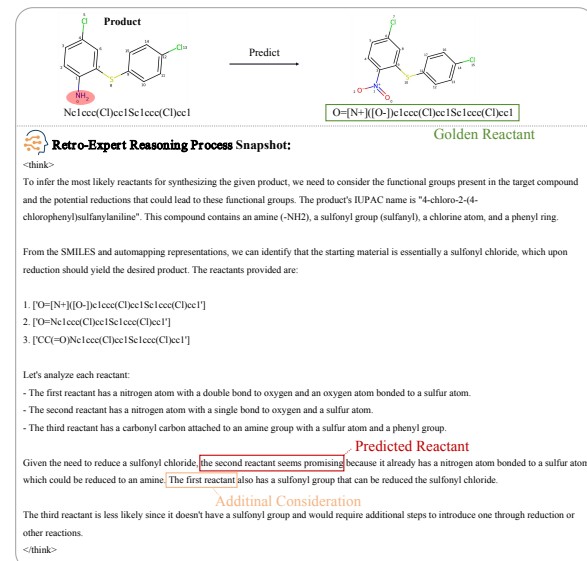

*(b)* Failure case for multiple possible reaction paths.

*Figure 17.* Visualization of failure case.

## G. Details of Comparison Methods

This section details all prompts employed in the LLM-based methods within our comparative experiments. Specifically, for chemical LLMs, which inherently adhere to prescribed predictive patterns. To ensure valid predictions, we strictly follow each model's native evaluation templates when generating output. The space-related information was incorporated in accordance with the respective evaluation templates.

For general LLMs, we employ a unified prompt structure identical to that used by our Retro-Expert, instructing the LLM to execute reactant predictions. Crucially, all general LLMs are provided with the identical chemical action space, comprising: (1) predicted reaction types, (2) predicted reaction centers, and (3) Top-4 reactant candidates generated by specialized models.

| |
|---|
| What are the possible reactants that could have formed the following product?
{Product Standard SMILES} |

2. The prompt for nach0 and ChemCrow.

| |
|---|
| There is a single choice question about chemistry. Answer the question by replying A, B, C, or D.

Question: What are the usual materials employed for creating {Product Standard SMILES}? Its automapping version is: {Product Mapped SMILES}. The IUPAC name of the product is: {IUPAC Name}.

A. {Top-1 Reactants Candidate}
B. {Top-2 Reactants Candidate}
C. {Top-3 Reactants Candidate}
D. {Top-4 Reactants Candidate}
Answer: |

3. The prompt for ChemLLM.

| |
|---|
| Chemical reaction equations are typically expressed in the following form:
`reactant1.reactant2.reactant3...>>product`. In this form, each substance (reactant or product) is represented using the SMILES notation.

Now we will provide you with an incomplete chemical reaction equation, where the missing part is represented by `"___"`. The missing part could consist of one or more substances.
Based on the remaining portions of the reaction equation, please infer what the missing part could be.

**Note:** Please only provide the missing part in your response, without any additional content.

**Incomplete equation:**
___ ≫ **{Product Standard SMILES}** |

4. The prompt for ChemDFM.

| |
|---|
| You are an experienced chemist analyzing chemical retrosynthesis.
Given the standard SMILES representation of the product is: {Product Standard SMILES},
Its automapping version is: {Product Mapped SMILES},
The IUPAC name of the product is: {IUPAC Name}.

The possible reaction type for synthesizing this product is: {Reaction Type}.
The possible reaction center for synthesizing this product is: {Reaction Center}.

**Reactants Candidate Set:**
(1) {Top-1 Reactants Candidate}
(2) {Top-2 Reactants Candidate}
(3) {Top-3 Reactants Candidate}
(4) {Top-4 Reactants Candidate}

Please reasonably infer the most likely reactants for synthesizing the product molecule from the Reactants Candidate Set.
Your response must strictly follow the **Output Format**. Do not include any explanation or additional text.
[Output Format]
`answer:  <the reactant SMILES>` |

5. The prompt for general LLM.

## H. Details of Out-of-Distribution Analysis

In this section, we provide the identical prompt to all benchmarked models under a uniformly chemical decision space.

---

There is a single choice question about chemistry. Answer the question by replying A, B, C, or D. The options are TopK predictions.

Question: What are the usual materials employed for creating: {Product Standard SMILES}? Please reasonably infer the most likely reactants for synthesizing the product molecule from the Reactants Candidate Set.

**Reactants Candidate Set**:
A. {Option A information in Chembench}
B. {Option B information in Chembench}
C. {Option C information in Chembench}
D. {Option D information in Chembench}

Your response must strictly follow the Output Format. Do not include any explanation or additional text.
[Output Format Example]:
```
answer:  B
```

---

9. The prompt for Out-of-Distribution Analysis.

## I. Limitations

Although Retro-Expert improves both retrosynthesis accuracy and reasoning interpretability, several limitations remain. First, our current model is trained only on a 7B backbone. Due to the limited local structural analysis ability and chemical knowledge of the 7B model, it may still make local chemical mistakes in the verbalized rationale, such as misidentifying functional groups or giving imperfect mechanistic descriptions, even when the final reactant decision is correct. Therefore, our claim is not that every generated rationale is fully chemically flawless, but that KGPO improves the model's candidate-level decision making, including selection, rejection, and self-correction, while providing a more interpretable reasoning pathway. In future work, we will explore stronger and larger LLM backbones to further improve local chemical analysis and reasoning fidelity.

Second, our extension to multi-step retrosynthesis is still preliminary. In this work, Retro-Expert performs step-wise reasoning over route-level candidates generated by specialized multi-step models. The current results indicate the potential of step-aware route reasoning beyond simple reranking, yet more systematic development is needed to handle harder multi-step planning scenarios. In future work, we will strengthen the framework for more complex multi-step planning and further validate more predicted routes through wet-lab experiments, aiming to discover more novel synthetic transformations.

