# OpenReview forum: "Retro-Expert: Collaborative Reasoning for Interpretable Retrosynthesis"
_ICML.cc/2026/Conference — ICML 2026 regular_

### Official Review · Reviewer_2B9i · 2026-03-05

**Soundness:** 2
**Presentation:** 3
**Significance:** 2
**Originality:** 2
**Overall Recommendation:** 4
**Confidence:** 4

**Summary:**

The authors propose "Retro-Expert", a hybrid retrosynthesis pipeline. It uses specialized models (like T5Chem and GraphRetro) to generate a bounded decision space containing reaction types, centers, and top-4 reactant candidates. An LLM (Qwen2.5-7B-Instruct) then acts as a router, evaluating this space to select or generate a candidate while outputting a chain-of-thought rationale. The model is fine-tuned using a reinforcement learning objective (KGPO) guided by a static, rule-based reward model (Chem-RM) that checks structural overlap and validity via RDKit.

**Compliance With Llm Reviewing Policy:**

Affirmed.

**Final Justification:**

The authors have addressed the primary technical and practical concerns raised in my initial review.
Given the strong engineering execution and the pragmatic approach to integrating LLMs into chemical workflows, I am raising my score to a **Weak Accept**.

**Key Questions For Authors:**

How do you guarantee Qwen2.5-7B-Instruct did not ingest the USPTO datasets or ChemBench targets during its 18T-token pre-training?

What is the wall-clock inference time (in milliseconds) for a single prediction? How do you intend to scale this approach for MSR, where MCTS requires thousands of SSR calls per target?

**Limitations:**

The authors adequately note failure cases involving molecules with multiple reactive sites.
Unaddressed: The paper completely ignores the massive computational cost and inference latency introduced by using a 7B parameter LLM for single-step prediction.

**Strengths And Weaknesses:**

### Strenghts
+ Constraining LLM generation using the outputs of specialized models is a mechanically sound approach to bound the action space and reduce hallucinations.
+ Employing RDKit for deterministic reward signals (Chem-RM) is a strong engineering choice. It avoids the known collapse issues of LLM-as-a-judge evaluation pipelines.
+ Validating two predicted routes in a physical chemistry lab is commendable and successfully demonstrates baseline physical viability.

### Weaknesses
- Data Contamination (LLM Pre-training Leakage): Qwen2.5-7B-Instruct was trained on roughly 18 trillion tokens. There is Insufficient Data in the paper verifying that the USPTO-50K, USPTO-FULL, and ChemBench target datasets were scrubbed from Qwen's pre-training or instruction-tuning corpus. Comparing a domain-finetuned model against zero-shot API baselines (Table 1) under potential data leakage invalidates the claim that the model is performing active "reasoning". It is highly probable the model is simply retrieving memorized priors.
- Single-Step Retrosynthesis (SSR) is only practically useful when integrated into Multi-Step Retrosynthesis (MSR) systems via search algorithms like Monte Carlo Tree Search (MCTS). These systems require thousands of SSR node expansions per target. Specialized models execute in milliseconds. Generating long-form reasoning tokens with a 7B parameter LLM for every MCTS node expansion is computationally prohibitive, making this architecture practically infeasible for high-throughput chemical routing.
- The authors claim the model exhibits "emergent self-reflection" when it rejects poor candidates. However, Equation 5 explicitly multiplies the RL advantage by a hardcoded reward ($r_{crit}$) that is granted only when the model rejects the small model's prediction. This is hardcoded incentive hacking, not emergent logic.
- Minor: Tables 2 and 3 omit competitive contemporary non-LLM models. Specifically, the paper excludes retroKNN and MHNreact among others.

---

> ### Author Rebuttal · Authors · 2026-03-31
>
> >**Q1: Beyond Memorization from Data Contamination.**
>
> We appreciate this concern. We cannot fully rule out the possibility that some relevant content is already contained in the Qwen2.5-7B-Instruct. However, we argue that the observed performance gains cannot be adequately explained by simply retrieving memorized priors alone.
>
>
> As shown in Table 1, the base model achieves only 0.02% accuracy when reasoning solely from the product. Even when provided with the same decision space, its accuracy rises to only 37.55%, which remains far below Retro-Expert’s 75.69%. Table 4 further shows that even with identical candidate information, other models still perform substantially worse than Retro-Expert (65.67%). These results suggest that the model is not merely retrieving memorized answers, but effectively uses the candidate information for chemical reasoning.
>
>
> In addition, we conducted a human evaluation to assess whether the model reaches correct answers through correct reasoning rather than simple memory retrieval. We randomly selected 500 samples and asked several chemistry PhDs in organic synthesis to rigorously evaluate the model’s reasoning process, focusing on whether the generated thinking reflected a chemically sound reasoning process toward the final predictions.
>
>
> |Answer|Reasoning|Coverage (%) |
> |-|-|-|
> |correct|chemically sound|82.85|
> |correct|partially flawed|17.15|
> |wrong|plausible|33.60|
> |wrong|flawed|66.40|
>
>
> >**Q2: Multi-Step Retrosynthesis.**
>
> We thank the reviewer for raising this important practical concern. In our multi-step setting, the LLM is not invoked at every single-step search node expansion, as in MCTS routing. Instead, we decouple route search from reasoning: the multi-step specialized model generates Top-K complete multi-step routes via beam search, and these routes are then jointly provided to the LLM in a single query as a route-level decision space. The LLM then performs a single reasoning (e.g. analyse, select, compare) over the candidate routes. This design assigns efficient search to specialized models and reserves high-value chemical decisions for the LLM, yielding improved accuracy at practical cost.
>
>
>
> >**Q3: Computational Cost.**
>
> We further report the computational cost of Retro-Expert on the retrosynthesis.
>
> |Setting|LLM|Specialized Models time / sample (s) ↓|LLM Reasoning time / sample (s) ↓|Total Latency / sample (s) ↓|
> |-|-|-|-|-|
> |single-step|ChemDFM |0.36|2.81|3.17|
> |single-step|ChemLLM |0.36|3.83|4.19|
> |single-step|Ours|0.36|3.26|3.62|
>
>
> |Setting|Method|Total Inference Time / sample (s) ↓|
> |-|-|-|
> |multi-step|DFPN|59.47|
> |multi-step|MCTS|51.73|
> |multi-step|Retro*|51.16|
> |multi-step|Ours|**20.41**|
>
>
> >**Q4: Self-Reflection.**
>
> We agree with the reviewer that the term ''emergent'' is not sufficiently precise. Equation (5) is designed to address the credit assignment problem in RL for retrosynthesis. In standard GRPO, advantages are computed through linear group normalization, which can underweight low-frequency yet high-value correction behaviors. By contrast, KGPO increases the relative advantage of trajectories that correct erroneous candidates, encouraging the LLM to evaluate candidates critically rather than follow them blindly. This mechanism does not hard-code rejection rules at the output level. Instead, it incorporates sparse yet important correction behaviors into explicit policy optimization. As shown in Table 6 & 7, the accuracy gains are accompanied by improved reasoning quality, indicating that KGPO promotes better reasoning trajectories rather than superficial rejection patterns. In addition, performance increases from 70.1% to 76.1% as more reactant candidates are provided, showing that the LLM makes effective use of candidate information during reasoning (in Table 10).
>
> We further analyze the model’s rejection behavior on the test set in below table. The result shows that KGPO exhibits a higher rejection rate for incorrect candidates while maintaining a high selection rate for correct ones, indicating that it can better discriminate candidate quality.
>
> |Setting|Candidate|Rejection Rate (%)|
> |-|-|-|
> |GRPO|Correct|0.02|
> |GRPO|Wrong|0.22|
> |KGPO|Correct|1.07|
> |KGPO|Wrong|**39.12**|
>
> >**Q5: Add More Baseline.**
>
> We thank the reviewer for this valuable suggestion. We will include RetroKNN and MHNreact in the revised Tables 2 and 3.
>
> |Dataset|Model|Top-1 Acc (%)| Top-3 Acc (%)|
> |-|-|-|-|
> |USPTO-50K|MHNreact|51.8|74.6|
> |USPTO-50K|RetroKNN|66.7|88.2|
> |USPTO-50K|Ours|75.7|90.8|

---

> > ### Author Rebuttal · Reviewer_2B9i · 2026-04-03
> >
> > The authors have addressed the primary technical and practical concerns raised in my initial review.
> > Given the strong engineering execution and the pragmatic approach to integrating LLMs into chemical workflows, I am raising my score to a **Weak Accept**.

---

> > > ### Author Response · Authors · 2026-04-05
> > >
> > > We sincerely thank the reviewer for the thoughtful suggestions and for supporting our work. We are especially encouraged by the reviewer's pragmatic perspective on integrating LLMs into chemical workflows, as well as on the value of chemical toolkit-based deterministic rewards. We also deeply appreciate the reviewer's concerns regarding data contamination, practical scalability, and the interpretation of self-reflection. These comments helped us substantially improve the paper: (1) clarify more carefully why the proposed method is not simply retrieving memorized priors; (2) provide a clearer description of the multi-step setting together with computational-cost analysis; (3) and revise the discussion of ''self-reflection'' to make KGPO's role in rejecting incorrect candidates clearer. We are truly grateful for the reviewer’s time and support.

---

### Official Review · Reviewer_wxeb · 2026-03-12

**Soundness:** 2
**Presentation:** 2
**Significance:** 2
**Originality:** 3
**Overall Recommendation:** 3
**Confidence:** 3

**Summary:**

This paper presents Retro-Expert, a framework for interpretable retrosynthesis prediction that combines specialized chemical models with an LLM reasoning engine trained via reinforcement learning. The core idea is to construct a "chemical decision space" using existing specialized models (reaction type classifiers, reaction center localizers, reactant generators), then have an LLM (Qwen2.5-7B-Instruct) reason over this space to select or generate reactant predictions while producing natural language explanations. The LLM is trained with a novel RL algorithm called Knowledge-Grounded Policy Optimization (KGPO), which extends GRPO with a non-linear advantage rectification that amplifies credit for correcting erroneous candidates. A training-free reward model (Chem-RM) provides dense rewards across three dimensions: structural analysis accuracy, reaction-related decision quality, and final answer correctness.

The authors evaluate on USPTO-50K and USPTO-FULL, reporting 75.7% Top-1 accuracy on USPTO-50K (surpassing the underlying Graph2Edits model at 67.2%) and 58.4% on USPTO-FULL. They also report an OOD evaluation on ChemBench (65.67%), interpretability assessments via GPT-4o and human experts, two wet-lab synthesis experiments (single reactions), and a preliminary multi-step retrosynthesis evaluation on PaRoutes. The paper claims this is the first retrosynthesis model capable of generating natural language reasoning processes.

**Compliance With Llm Reviewing Policy:**

Affirmed.

**Final Justification:**

The flawed reasoning examples and the initial overclaims/inflated language lead me to maintain my score.

**Key Questions For Authors:**

1. **Reasoning fidelity:** In several visualized examples (Figures 12, 13), the model's reasoning contains clear chemical errors (e.g., misidentifying functional groups, incorrect mechanistic descriptions) yet still arrives at the correct answer. How do you reconcile this with the claim of "genuine chemical reasoning"? What fraction of correct predictions are accompanied by chemically sound reasoning (not just high LC/MA scores from GPT-4o)? A targeted analysis correlating answer correctness with reasoning correctness — verified by human experts, not GPT-4o — would substantially strengthen or weaken the interpretability claims.

2. **Isolating KGPO's contribution:** Can you provide a comparison of GRPO vs. KGPO using the identical full Chem-RM reward (i.e., rbase + rreact + rstruct)? The current Table 7 comparison uses rbase-only for GRPO, making it impossible to isolate the effect of the non-linear advantage rectification from the effect of richer rewards.

3. **Multi-step evaluation details:** How exactly is Retro-Expert adapted for multi-step retrosynthesis on PaRoutes? What search algorithm is used? Is the LLM called at each step? Are reasoning traces generated per step? What is the computational cost compared to baselines? The current description is insufficient to evaluate this contribution.

4. **Decision space coverage:** What fraction of the USPTO-50K test set has the correct answer present in the Top-4 candidate reactants? This upper-bound on selection accuracy is critical context for interpreting the 75.7% result. Similarly, what fraction of the test set falls into the "all candidates incorrect" category where self-generation is needed?

5. **Training stability and data sensitivity:** You train on only 9K samples from USPTO-50K. How sensitive is performance to the specific training subset? Can you report variance across multiple training runs or different random samples? The absence of any uncertainty quantification is a notable gap.

**Limitations:**

- The reasoning traces, while improved over baselines, still contain frequent chemical errors, limiting their trustworthiness for expert decision-making.
- The framework's performance is fundamentally bounded by the coverage of the specialized models' candidate sets.
- Single-step retrosynthesis, the primary evaluation setting, has limited direct practical utility compared to multi-step route planning.
- The computational overhead of LLM inference at each retrosynthetic step is not discussed — this could be prohibitive for iterative multi-step planning.

**Strengths And Weaknesses:**

### Strengths

- **Practically relevant framing.** The motivation — that existing retrosynthesis models are black boxes and chemists need interpretable reasoning — is well-grounded and addresses a real barrier to adoption. Providing natural language explanations alongside predictions is a meaningful contribution to usability.

- **Plug-and-play modularity.** The framework's model-agnostic design, where any retrosynthesis model can be swapped in at inference without retraining the LLM, is a genuine practical advantage. Table 2 demonstrates this across six different specialized models, showing consistent improvements over each baseline.

- **Emergent self-correction capability.** The finding that Retro-Expert can generate correct answers even when all provided candidates are incorrect (43.1% success rate in such cases, Figure 3) is a compelling emergent behavior. This moves the system beyond a simple re-ranker.

- **Reward model design.** Chem-RM is a reasonable approach to providing dense, training-free rewards for chemical reasoning. Using RDKit-based structural verification to ground rewards in objective chemical properties avoids the hallucination problems of learned reward models.

### Weaknesses

- **Overclaimed contributions and inflated language.** The paper's rhetoric significantly overstates the novelty and impact of several components. Claiming this is "the first retrosynthesis model capable of generating natural language interpretable reasoning processes" ignores recent work (e.g., Bran et al., 2025, which is cited but insufficiently discussed) that also produces reasoning for retrosynthesis via LLMs. The repeated framing of the LLM as performing "deep reasoning" and "logical deduction" is not well-supported — the visualized reasoning traces (Appendix E) frequently show the model arriving at correct answers through superficial pattern matching or even chemically incorrect logic (see Figures 12 and 13, where the reasoning contains factual errors yet the correct candidate is still selected). The paper would benefit from more measured claims and honest acknowledgment that the reasoning quality, while improved, remains far from expert-level.

- **The task is primarily single-step retrosynthesis, which limits practical relevance.** The vast majority of experiments focus on single-step retrosynthesis prediction on USPTO-50K and USPTO-FULL. In practice, chemists need multi-step route planning. The multi-step evaluation (Section 4.7, Table 8) is minimal: only Top-1 accuracy on PaRoutes is reported with no description of how the framework is adapted for multi-step planning, what search algorithm is used, what the decision space looks like in this setting, or whether reasoning traces are generated at each step. This section reads as an afterthought rather than a thorough evaluation.

- **The comparison in Table 1 is misleading.** Retro-Expert receives Top-4 reactant candidates plus reaction type and center predictions from specialized models (how are the reaction type and center obtained from only product information?), while the general LLM baselines in the "without space" setting receive none of this. The 75.69% vs. ~0–4% comparison is therefore uninformative — it compares systems with fundamentally different information access. Even in the "with space" setting, Retro-Expert benefits from RL training while the general LLMs are used zero-shot. A fairer comparison would be to fine-tune or few-shot the general LLMs under the same decision space.

- **The improvement over specialized models is bounded by the decision space.** Table 2 shows that Retro-Expert improves Top-1 accuracy over the underlying model by 3–8 percentage points, but Top-10 accuracy improvements are marginal or nonexistent (e.g., 96.3 → 96.7 for LocalRetro). This suggests the framework primarily re-ranks and occasionally corrects candidates but cannot fundamentally exceed the coverage of the specialized models. The 43.1% self-generation success rate (Figure 3) applies only to the subset where all candidates are wrong — what fraction of the total test set does this represent? This context is missing.

- **Reasoning quality evaluation is unconvincing.** The interpretability metrics (MA, FC, LC) rely heavily on GPT-4o as an automated judge, which is problematic for evaluating chemical correctness. The human evaluation involves only three experts and is reported as averages without inter-annotator agreement, sample sizes, or confidence intervals. More critically, inspecting the visualized reasoning examples in the appendix reveals concerning patterns: in Figure 12 (heteroatom alkylation), the model's reasoning about sulfonyl chlorides and carbamate formation is chemically incoherent, yet it arrives at the correct answer. In Figure 13 (C-C bond formation), the reasoning about atom positions and functional group assignments contains clear factual errors. These examples undermine the claim that the model performs "genuine chemical reasoning" — it appears to often select correctly despite flawed reasoning, suggesting the reasoning trace may be more post-hoc rationalization than actual decision logic.

- **Wet-lab validation is extremely limited.** Two reactions — a Suzuki coupling and a Jones oxidation — are both well-established reaction types. Neither represents a challenging or novel synthetic transformation. The claim that the Suzuki coupling constitutes the "first-ever reported synthesis" of 3-(2-ethoxyphenyl)thiophene is technically about the specific compound lacking a documented route, not about any novel chemistry. A more convincing validation would involve a structurally complex target or an unusual disconnection strategy. Additionally, there is no discussion of the yield optimization, side products, or how many predicted routes were attempted before finding successful ones.

- **KGPO's advantage over GRPO is not clearly isolated.** The non-linear rectification in Equation (5) adds a multiplicative factor based on critical correction rewards. However, the comparison in Table 7 between GRPO (66.8%) and KGPO (75.7%) conflates the effect of the non-linear advantage with the full Chem-RM reward signal. The GRPO baseline uses only `r_base`, while KGPO uses the complete Chem-RM. To isolate KGPO's contribution, one would need GRPO with the full Chem-RM reward versus KGPO with the full Chem-RM reward. This comparison appears to be partially addressed in Table 6 (rbase + rreact = 72.3% vs. Ours = 75.7%), but it remains unclear whether the 72.3% uses KGPO or GRPO optimization.

- **Training data is very small and curation is under-described.** Only 9K samples from USPTO-50K are used for training (from a ~40K training set). The stratified sampling strategy based on template frequency tiers is described briefly but raises questions: how sensitive is performance to the specific training subset? What happens with a different random sample? No variance across training runs is reported.

- **Presentation issues.**
  - The paper contains several grammatical errors and typos throughout.
  - The term "chemical decision space" is used extensively but is essentially just the Cartesian product of candidate sets from specialized models — the terminology is unnecessarily grandiose.
  - The reward coefficients (λ₁=1.5, λ₂=2.0, λ₃=1.0) are presented without justification or sensitivity analysis.

---

> ### Author Rebuttal · Authors · 2026-03-31
>
> >**Q1: Overclaims and Inflated Language**
>
> We appreciate the insightful comment. We will revise the overclaims and imprecise terms (e.g., deep reasoning) throughout the paper. Chemical decision space denotes chemical knowledge anchors that alleviate the LLM’s cold-start problem in retrosynthesis. Our work uses an LLM to reason over candidates (select, reject, correct), producing both rationale and predictions. Bran et al. (2025) use an LLM to analyze and score a given route against an expert query.
> >**Q2: Reasoning Fidelity**
>
> We accept that our claims about reasoning quality should be more precise. Our method aims to improve the LLM's decision-making (select, reject, correct) over candidates, while producing an interpretable pathway. We agree that Figures 12/13 reveal an important limitation. The model can recover the correct decisions while still making local chemical mistakes in the verbalized rationale. This issue is primarily due to the 7B model's limited ability in local structural analysis and chemical knowledge. Notably, the condensed rationale shows only part of the LLM's reactant-level decision.
>
> We conducted a human evaluation on 500 randomly sampled cases by some PhDs in organic chemistry (Inter-rater $\kappa$ = 0.78). The results suggest that most correct predictions are grounded in valid chemical logic rather than post-hoc rationalization. We will revise the wording ''genuine chemical reasoning'' accordingly and clarify the limitations.
> Answer|Reasoning|Coverage (%)
> -|-|-
> correct|chemically sound|82.85
> correct|partially flawed|17.15
> wrong|plausible|33.60
> wrong|flawed|66.40
> >**Q3: Multi-Step Evaluation**
>
> We appreciate this important issue. In our multi-step setting, the LLM is not invoked at every single-step search node expansion, as in MCTS routing. Instead, we decouple route search from reasoning: the multi-step specialized model generates Top-K complete multi-step routes via beam search, and these routes are then jointly provided to the LLM in a single query as a route-level decision space. The LLM then performs a single round of reasoning (e.g., analyze, select, compare) over the candidate routes. This design assigns efficient search to specialized models and reserves high-value chemical decisions for the LLM, yielding improved accuracy at practical cost. The computation cost is shown below, More results can be seen in Reviewer-baRX Q1.
> Method|Inference Time / sample (s)↓
> -|-
> DFPN|59.47
> MCTS|51.73
> Retro*|51.16
> Ours|20.41
> >**Q4: LLM Baseline Comparison**
>
> We compare training strategies on the same backbone (Qwen2.5-7B-Instruct) with identical decision space (i.e.,type, center, reactants). Types and centers are predicted from the product alone via T5Chem and GraphRetro (Lines 311-314).
> Space Setting|Training Strategy|Top-1 Acc (%)
> -|-|-
> w/ space|Zero-shot|37.55
> w/ space|Few-shot|40.67
> w/ space|SFT|58.92
> w/ space|GRPO|66.82
> w/ space|KGPO|75.69
> >**Q5: Decision Space Coverage & Self-Reflection**
>
> We acknowledge that the main gain comes from selecting and correcting the candidates. In our framework, self-reflection is not a hard-coded procedure for the ''all candidates incorrect'' case, but a general reasoning policy applied to every sample. The LLM analyzes the candidates and then autonomously decides whether to select, reject, or generate a new answer, while producing an interpretable pathway. Figure 3 shows that it generates new correct answers for 1.7% of all samples, while the ''all candidates incorrect'' subset accounts for about 4% of the test set, suggesting the LLM can partially extend beyond the coverage of the specialized models.
> >**Q6: Isolating KGPO's Contribution**
>
> We compare GRPO and KGPO under different reward settings. The self-correction rate is the proportion of test samples where the model generates a new correct answer when the candidates are incorrect.
> Setting|Reward Model|Top-1 Acc (%)|Self-Correction Rate (%)
> -|-|-|-
> GRPO|$r_{base}$|66.8|0.03
> GRPO|Chem-RM|73.1|0.91
> KGPO|$r_{base}$|66.8|0.03
> KGPO|Chem-RM|75.7|1.73
> >**Q7: Wet-Lab Validation**
>
> We further tested our model on more challenging targets. For 3-oxa[3.1.1]propellane and 3-aza[3.1.1]propellane, our predicted routes are consistent with a paper published in Nature Chemistry (Feb 25, 2026). This demonstrates that our approach can identify novel synthetic transformations.
>
> The cases in paper are the LLM’s Top-1 answers, with yields and detailed procedures provided in Appendix D. The wet-lab case selection is detailed in Reviewer GruX-Q3.
> >**Q8: Training Stability & Data Sensitivity**
>
> We add experiments on data sensitivity and reward weights sensitivity. The results show that higher quality, more balanced 9k subsets better support reasoning strategy learning. We prioritize $r_{react}$ to optimize decision-making during reasoning.
> Training Data|Top-1 Acc. Mean(%)|Var.
> -|-|-
> 6k|71.35|0.06
> 9k|75.69|0.04
>
> $\lambda_1$|$\lambda_2$|$\lambda_3$|Top-1 Acc. (%)
> -|-|-|-
> 1.5|2.0|1.0|75.7
> 1.5|1.0|1.0|75.1
> 1.5|3.0|1.0|75.8

---

> > ### Author Rebuttal · Reviewer_wxeb · 2026-04-02
> >
> > Thanks to the authors for their rebuttal, there were improvements, but some major weaknesses remain:
> > - [Q2] The new data is valuable, but the framing tries to minimize a real problem. I'm not sure what to think of "condensed rationale shows only part of the LLM's reactant-level decision", which sounds like the full reasoning is better than what was shown in the manuscript.
> > - [Q3] Multi-step evaluation is a lot clearer. It seems the approach is based on route reranking rather than step-by-step reasoning, which makes it less exciting.
> > - [Q6] GRPO vs KGPO, when the same Chem-RM reward is used, is only a 2.6 point gain (and not the 8.9 point gain that was shown in the original table)

---

> > > ### Author Response · Authors · 2026-04-05
> > >
> > > >**Q2: Reasoning Fidelity.**
> > >
> > > We appreciate the reviewer's comment. We regret that our previous wording was not precise enough and could be read as implying that the unshown full reasoning is better than what appears in the manuscript. That is not our intent.
> > >
> > > Our point was only that the visualized examples showed a summary of the reasoning trace, which did not fully include the candidate analysis (e.g., type, center). This was a presentation choice, not a claim that the omitted parts are better or that they would resolve the visible limitation. To be clear, the visualized examples reflect an important limitation of the current 7B model: it can make the correct decisions (i.e., type, center, reactants) while still producing local chemical mistakes in the verbalized rationale, likely due to its limited fine-grained structural analysis and chemical knowledge. We will revise the wording accordingly and clarify this important limitation.
> > >
> > > >**Q3: Beyond Reranking by Step-Aware Reasoning.**
> > >
> > > We respectfully clarify that **our model is not limited to route reranking**. Beyond reranking, it can also **exploit partially correct local decisions across different routes and combine them into a better final answer** (e.g., using one Step-2 branch from Route-A and another Step-2 branch from Route-B to form the Step-2 in the new route). **It can also generate a new route** conditioned on the candidate routes when it finds none of them satisfactory. This behavior goes beyond conventional reranking, which typically only scores the complete route as a whole.
> > >
> > > For example, in the synthesis of 2-phenylethyl benzoate (C1=CC=C(C=C1)CCOC(=O)C2=CC=CC=C2), our model keeps Route-A's Step-2 branch to produce 2-phenylethanol (c1ccc(cc1)CCO) from ethylene oxide (C1CO1) and phenylmagnesium bromide (Br[Mg]c1ccccc1), while adopting Route-B's Step-2 branch to obtain benzoic acid (c1ccccc1C(=O)O) from toluene (Cc1ccccc1) via potassium permanganate oxidation. It rejects Route-A's benzoic-acid branch as more demanding and complex, stating: ''This step demands harsh reaction conditions due to strong acid catalysis...It appears more complex than Route-B for the preparation of benzoic acid.''
> > >
> > >
> > > This behavior is mainly enabled by **step-aware reasoning**. Specifically, our model reasons at two levels. At the step level, it examines whether the transformation in each step is chemically plausible and whether the predicted intermediate is consistent with the proposed reactants and reaction logic. At the route level, it compares routes based on step-level analysis. For example, ''I rule out Route-A because its Step-2 product lacks the aldehyde required for the subsequent oxidation.''
> > >
> > >
> > > >**Q6: KGPO's Full Gain.**
> > >
> > > We thank the reviewer for this important comment. In our paper, **KGPO denotes the full framework consisting of both Chem-RM (in Eq. (3)) and the rectified policy optimization algorithm (in Eq. (5))**, whereas GRPO uses only the final-answer reward ($r_{base}$). To make this distinction explicit, we reorganize the table as follows. This decomposition clarifies that **the 8.9-point gain reflects the full-system gain of KGPO over GRPO**, whereas **the 2.6-point gain reflects the gain of the rectified update of KGPO**. We also note that a 2.6-point gain is meaningful at this performance level, given the strength of the baseline. This suggests that the rectified update provides a real benefit by encouraging the model to critically reject incorrect candidates rather than follow them blindly.
> > >
> > > | Configuration | Top-1 Acc. (%) | Self-Correction Rate (%) |
> > > |---|---:|---:|
> > > | GRPO ($r_{base}$) | 66.8 | 0.03 |
> > > | GRPO + Chem-RM | 73.1 | 0.91 |
> > > | KGPO (Chem-RM + rectified update) | 75.7 | 1.73 |
> > >
> > > Importantly, KGPO contributes in two ways: (1) Reward Modeling (in Eq. (3)): we introduce Chem-RM, a reward model that utilizes the chemical toolkit (e.g., RDKit) to provide dense and accurate reward signals grounded in intrinsic deterministic chemical knowledge. It avoids the known collapse issues of LLM-as-a-judge evaluation pipelines. (2) Algorithm Optimization (in Eq. (5)): standard GRPO computes advantages via linear group normalization, which can underweight low-frequency yet high-value correction behaviors. By contrast, KGPO increases the relative advantage of trajectories that correct erroneous candidates, encouraging the LLM to evaluate candidates critically rather than follow them blindly. We thank the reviewer again for prompting us to clarify this distinction.
> > >
> > >
> > > We sincerely hope the reviewer will consider the overall contribution and empirical gains of the work, in light of its practical framing, self-correction capability, and reward model design recognized in the review, which are also reflected in the broader review feedback on its pragmatic significance, strong engineering execution, and value for producing high-quality data.

---

### Official Review · Reviewer_baRX · 2026-03-13

**Soundness:** 3
**Presentation:** 2
**Significance:** 3
**Originality:** 4
**Overall Recommendation:** 5
**Confidence:** 5

**Summary:**

They propose using domain models that were trained on retrosynthesis planning as “knowledge anchors” for LLMs. The outputs of the domain-specific models are passed to the LLMs; in this way, LLMs have to first reason about the different proposals that they receive, and then choose what they think is the best. They believe this makes the LLM more interpretable.

Then they propose to use this configuration in an RL environment, where they reward using the proposed function to score model completions based on whether the completions contain reasoning about the different subgraphs of the molecules, the different components of the reaction, and the correctness of the proposed solution.

Applying this to the USPTO dataset, they find that the trained model greatly improves the frontier and chemical LLMs on the same task and in other chemistry benchmarks. The method also shows improvements over the domain-specific models.

**Compliance With Llm Reviewing Policy:**

Affirmed.

**Final Justification:**

I share the concerns raised by the other reviewers regarding the model itself. However, I find the authors’ proposed reinforcement learning method quite interesting (substantially better than distilling from closed‑source LLMs or simply scoring the final outcome of the full route). As such, it represents a valuable addition to the community. The authors demonstrate that their method works, with technically robust and solid results.

In response to the reviews, the authors have promised to rephrase parts of the paper to focus more on the method and less on the model. I am not entirely comfortable raising the score based on a promise alone, but I believe the method can still be extracted from the work as it currently stands.

To reiterate: the intelligent way they propose for distilling knowledge from domain‑specific models into LLMs is novel, interesting, and well executed. I therefore raise my score to Accept.

**Key Questions For Authors:**

- Why not improve or target multi-step training more?
- Did the authors check how the LLM was analyzing domain models' outputs, or did they simply trust the reward to shape this?
    - If they simply trusted the reward, I would be very curious to see how this changed as the training evolved.
- Why only 2-domain model outputs for training?
- Does scaling the model yield better results? Right now the model trained is quite small (only ~7B)?

I will be happy to increase the score if some of the comments above are addressed.

**Limitations:**

There are no limitations discussed in the current main text of the manuscript

**Strengths And Weaknesses:**

## Strengths

- It is a really nice idea that combines the best of both worlds, the domain-specific models accuracy, and the LLMs reasoning capabilities.
- The execution is correct as the results show improvement over the different benchmarks and the other evaluations that are presented in the work.
- The presentation and soundness of the paper are nice, with an easy-to-read drive through all the carried out experiments.

## Weaknesses

- I miss more analysis, results, and discussion on the multi-step retrosynthesis. What is shown in Table 8 is promising, but I believe that it should be the main results of the paper. At the end, real-world retrosynthesis is most of the time multi-step.
- I think it would be fair to reference some previous works, such as arXiv:2503.08537. They do not train, but they provide expert-based annotations to base models, showing that with these models, they can serve as great assistants.
- I am not sure if using LLMs, which cannot even write proper SMILES, is a good second solution when there is no plausible candidate.
    - Also, if this is a one-step process, how did the authors ensure that the LLM always considers the given candidates first and does not trust its internal knowledge from the beginning?
    - Did the authors evaluate this somehow?
    - Did the authors implement some solution to avoid this, so that the model simply answers without “considering” the knowledge it receives, or do they mostly trust the reward and RL to shape this?
- This is more like a comment, but I would love to see an evaluation of the most recent Gemini-Pro and Claude on these tasks. I have heard that they have become quite good at these things.
- How did the authors arrive at the use of 2 domain-model outputs for training?
    - Is this some kind of tuning, or was the decision arbitrary?
    - Also, why is only one retrosynthesis model output used in inference? Why not use all of the available ones?

---

> ### Author Rebuttal · Authors · 2026-03-31
>
> >**Q1: Multi-Step Retrosynthesis**
>
> We agree with the reviewer on the importance of multi-step retrosynthesis. In our multi-step setting, the LLM is not invoked at every single-step node expansion, as in MCTS routing. Instead, we decouple route search from reasoning: the multi-step specialized model (DMS Model) generates Top-K complete multi-step routes via beam search, and these routes are then jointly provided to the LLM in a single query as a route-level decision space. The LLM then performs a single round of reasoning (e.g. analyze, select, compare) over the candidate routes. This design assigns efficient search to specialized models and reserves high-value chemical decisions for the LLM, yielding improved accuracy at practical cost.
>
> We provide the results in the table below. The LLM is given four candidate routes, and the multi-step retrosynthesis model is run with a beam size of 50. During inference, we observe that the LLM analyzes individual steps within each candidate route and compares routes globally, indicating both step and route-level reasoning. For example, ''I rule out Route-1 because its Step-2 product lacks the aldehyde required for the subsequent oxidation.'' We will further improve the model’s reasoning for multi-step retrosynthesis.
>
> Method|Success Rate (%)|Avg. Route Length|Top-1 Acc (%)|Top-5 Acc (%)|Inference Time/sample (s) ↓
> -|-|-|-|-|-
> DFPN|84.75|3.2|11|17|59.47
> MCTS|97.14|4.8|17|46|51.73
> Retro*|97.26|3.5|15|41|51.16
> Ours|98.39|3.4|41|54|20.41
>
>
> >**Q2: More Work**
>
> We appreciate the work (arXiv:2503.08537) on LLM-based molecular analysis and synthesis planning, which has greatly inspired our study. We will certainly cite it and elaborate further.
>
>
> >**Q3: Improving LLM-Generated SMILES Validity**
>
> Thank you for this comment. General LLMs often struggle to generate valid SMILES reliably. In our framework, specialized models provide candidate sets as knowledge anchors, which increase the validity of LLM generated SMILES to over 90% (Table 1, Val.). Even when all candidates are wrong, the LLM can still make effective corrections on them, partially overcoming the limitations of the input specialized models (in Figure 3).
>
>
> >**Q4: The LLM Considers the Candidates First**
>
> To address this concern, we first explicitly present the candidates in the prompt, providing the LLM with a natural starting point for reasoning. Second, we employ the reward model with RL to encourage the LLM's correct decisions (e.g. select, reject, self-generate) over the candidates. In Table 9 & 10, performance improves by adding type/center cues or increasing reactant candidates. Moreover, even when all reactant candidates are wrong, the LLM can still recover a correction from the candidates and its internal knowledge (in Figure 3). This suggests that the LLM makes effective use of these candidates.
>
> We also compare reasoning traces before and after RL, indicating the increased use of candidates with RL training. The result suggests the LLM first tends to incorporate candidates in reasoning.
>
>
> Stage|Candidates Analysis Rate (%)|Rejection Rate (%)
> -|-|-
> Before RL|68.5|0.04
> After RL|95.7|4.87
>
>
> >**Q5: Add Baselines**
>
> We appreciate the feedback and add the baseline based on the same decision space, which provide the same candidates (i.e., reaction types, center, reactants).
>
> Model|Top-1 (%)
> -|-
> Gemini-2.5-Pro|50.6
> Claude|55.7
> Ours|75.7
>
>
> >**Q6: Two Domain-Model Outputs for Training**
>
> The content of the decision space was determined in consultation with chemistry experts. In retrosynthetic analysis, chemists typically first infer a plausible reaction family (type), then identify the key reactive site (center), and finally instantiate specific precursor candidates. Accordingly, we use two specialized models that produce three types of outputs (i.e. reaction type, center, reactants) to construct a decision space. As type and center information is gradually introduced, accuracy improves from 29.1% to 75.7% (in Table 9), which is consistent with human decision process.
>
>
> >**Q7: Combining Multiple Retrosynthesis Models**
>
> Thank you for this valuable suggestion. Our work primarily focuses on enhancing the LLM’s chemical reasoning ability. Table 2 shows consistent gains across different models, confirming the effectiveness of LLM-based reasoning.
>
> It is a nice idea to use the outputs of more retrosynthesis models to enhance predictive accuracy. To this end, we collect the Top-4 predictions from three models. Then the LLM reasons over the merged candidates. The results are shown below.
>
> Ensemble Model|Top-1 (%)
> -|-
> Ours (GraphRetro)|70.1
> Ours (LocalRetro)|70.7
> Ours (Graph2Edits)|75.7
> Ours (GraphRetro + LocalRetro)|71.7
> Ours (GraphRetro + Graph2Edits)|76.4
> Ours (LocalRetro + Graph2Edits)|76.0
> Ours (All)|76.9
>
> >**Q8: Scaling the Model**
>
> We conducted experiments by increasing the model from 3B to 7B raised Top-1 accuracy from 66.81\% to 75.69\%. We will add results with larger models.

---

> > ### Author Rebuttal · Reviewer_baRX · 2026-04-02
> >
> > Some of my concerns are addressed (e.g., the multi-step retro concerns, authors should consider using the same level of detail as that table in the other reviewer responses :), not only in mine).
> >
> > That said, I think my comments are partially unresolved because the authors use or even "abuse" the expert argument. And for some of the cases it is ok I guess, like about the two-model used for training (sounds too fancy but I guess it has sense). But then, in things like the reasoning reliability that other reviewers pointed to, the expert argument is a cheap argument, which, in my opinion, loses value if there is no annotation analysis. With this annotation analysis, I mean studying when the experts agreed with each other, for example, (some high-quality examples are in the GPQA manuscript), or explaining what the details for those annotations were.
> >
> > In addition, the argument about the LLM considering the candidates all the time is not very convincing. Even the strongest models hallucinate, so the ~7B ones probably do it more, and this work simply relies on their trust in the loss function without analyzing if it is working. Do not read me wrong, I think the loss function makes a lot of sense, but I think analysis is needed.
> >
> > On a final note, I would like to ask the authors to seriously consider rephrasing the history of their work into how valid the method is to generate good-quality data. I think this method is much cheaper than distilling from closed-source LLMs (that I know some labs and academic groups are doing), and I think this is partially why I scored higher than the other reviewers. I see this work as a method for producing high-quality data, demonstrated by the results of the trained models. Since the paper is framed around the model, some concerns that other reviewers pointed to, such as the evaluation being focused only on USPTO, are very valid.
> >
> > Also, related to the point above, I would encourage opening the data (i.e., by uploading it to HF), as I think it is high-quality data, or at least much better than the stuff currently out there for natural text reaction data.

---

> > > ### Author Response · Authors · 2026-04-05
> > >
> > > > **Q1: Annotation Analysis of Human Evaluation.**
> > >
> > > We thank the reviewer for this valuable feedback. We agree and provide more details.
> > >
> > > We revisited the annotations for 500 samples in the human evaluation. Each case was independently assessed by three PhDs in organic synthesis. The annotators were given the reasoning process produced by our model together with the ground-truth reaction information. To facilitate inspection, we also supplied molecular visualizations of the product and candidate reactants. We asked them to evaluate the reasoning process from multiple dimensions and provide a brief written justification for each dimension judgment. For example, ''Incorrect. The reacting functional group in the product is an aryl bromide, not an alcohol.''
> > >
> > > We annotated each case along six dimensions: (1) Answer Correctness (correct / incorrect), (2) Product Analysis Correctness (correct / partially flawed / incorrect), (3) Candidate Decision Correctness (correct / incorrect / not explicit), (4) Decision Rationale Correctness (reasonable / partially flawed / unreasonable ), (5) Reasoning Faithfulness  (faithful / weakly faithful / unfaithful), and (6) Annotator Confidence (high / low).
> > > A case is counted as **overall correct reasoning** only if all six dimensions are judged as correct, faithful, or high-confidence by at least 2 of the 3 annotators, making this a highly stringent criterion.
> > >
> > >
> > > The results are shown in the table below.
> > >
> > > | Annotation Dimension | 3/3 Agreement (%) | $\geq$ 2/3 Agreement (%) |
> > > |---|---:|---:|
> > > | Product Analysis Correctness | 80 | 99 |
> > > | Candidate Decision Correctness (type, center, reactants) | 77, 85, 79 | 100, 100, 100 |
> > > | Decision Rationale Correctness (type, center, reactants) | 73, 78, 71 | 98, 98, 96 |
> > > | Reasoning Faithfulness | 83 | 98 |
> > > | Overall Correct Reasoning | 59 | 63 |
> > >
> > >
> > > Following GPQA, **we further analyzed cases where the experts showed high agreement with each other** on the reasoning faithfulness dimension. Overall, the 3/3 unanimous agreement rate is 83%, and the $\geq$ 2/3 majority agreement rate is 98%. We then provide a more detailed analysis of 3/3 agreed cases.
> > >
> > > | Category | Percentage (%) |
> > > |---|---:|
> > > | Fully sound, well-supported reasoning with an answer that matches the ground truth. | 57 |
> > > | Fully sound, well-supported reasoning with an answer that does not match the ground truth (i.e., experts judged it to be a chemically plausible alternative). | 6 |
> > > | Largely sound but partially flawed reasoning with an answer that matches the ground truth. | 16 |
> > > | Weakly supported reasoning with an answer that matches the ground truth. | 2 |
> > > | Both incorrect answer and reasoning. | 19 |
> > >
> > > We agree that this kind of annotation analysis is more informative and valuable than invoking expert judgment alone, although it is still limited at present. We will add the annotation protocol & statistics, and representative high-agreement and contested examples in the revision.
> > >
> > >
> > > > **Q2: The LLM Considers the Candidates.**
> > >
> > > Thank you for this thoughtful comment. We acknowledge that hallucination and candidate bypass still occur in some cases. In our previous rebuttal, 95.7% of the test cases involved full analysis of the candidate set by our model. Our reward is intended to encourage the model to analyze candidates, select the correct ones, and reject & self-correct incorrect ones. Following the reviewer’s suggestion, **we further analyze whether the learned policy is actually working** during inference. We provide more details in below table.
> > >
> > >
> > > | Candidate-Grounded Analysis | LLM Behavior | Behavior Coverage (%) | Correct Predictions via this Behavior (%) |
> > > |---|---|---:|---:|
> > > | Yes (**95.71%**) | Select from candidates through full analysis | 90.84 | 71.64 |
> > > | Yes (**95.71%**) | Reject and self-correct from candidates through full analysis | 4.87 | 1.73 |
> > > | No (4.29%) | Select from candidates through superficial analysis | 3.93 | 2.28 |
> > > | No (4.29%) | Off-space prediction that bypasses candidates | 0.36 | 0.04 |
> > > | Total (100%) | - | 100.00 | **75.69** |
> > >
> > > > **Q3: Data-Centric Framing.**
> > >
> > > We appreciate the reviewer’s insightful perspective that Retro-Expert can also be framed as a practical pipeline for generating higher-quality data. We will revise the paper accordingly to place greater emphasis on this aspect. In addition, we will certainly release the relevant data publicly.
> > >
> > > Thank you for the time you put into reviewing our work. Your comments have genuinely improved this paper.

---

### Official Review · Reviewer_GruX · 2026-03-13

**Soundness:** 2
**Presentation:** 3
**Significance:** 3
**Originality:** 2
**Overall Recommendation:** 3
**Confidence:** 3

**Summary:**

The paper proposes a collaborative retrosynthesis prediction framework called Retro-Expert. Its core innovation lies in integrating a "chemical decision space," constructed by specialized chemical models (e.g., reaction type classifiers, reaction center localizers), with the deep reasoning capabilities of large language models (LLMs). Through a reinforcement learning approach termed Knowledge-Grounded Policy Optimization (KGPO), the framework not only predicts reactants but also generates human-interpretable reasoning processes in natural language. Experiments conducted on the USPTO-50K and USPTO-FULL datasets demonstrate its predictive accuracy, interpretability, and ability to validate novel synthetic pathways through wet-lab experiments.

**Compliance With Llm Reviewing Policy:**

Affirmed.

**Key Questions For Authors:**

See Weaknesses

**Limitations:**

yes

**Strengths And Weaknesses:**

* Strengths

(1) The experimental design is comprehensive, incorporating not only quantitative comparisons on standard datasets but also ablation studies, human evaluations of interpretability, and novel wet-lab validations, all of which enhance the credibility of the results.

(2) Integrating the reasoning capabilities of LLMs with domain-specific knowledge from specialized models to achieve interpretable retrosynthesis prediction represents an important and valuable research direction.

* Weaknesses

(1) The analysis of reasoning reliability is insufficient. The model might arrive at a correct answer by chance while generating a plausible-sounding, yet unfaithful, explanation.

(2) The paper lacks any discussion of computational efficiency or deployment feasibility. The proposed Retro-Expert framework requires multiple specialized models plus a 7B-parameter LLM fine-tuned via RL with extensive generation lengths, yet provides no analysis of inference latency, training cost, or scalability. This omission makes it difficult to assess whether the framework's pursuit of interpretability and accuracy comes at the expense of practical utility in real-world applications.

(3) The wet-lab validation, while a valuable addition, raises questions about its connection to the model's reasoning process. These experiments confirm the feasibility of the proposed synthetic routes but do not substantiate the correctness of the model reasoning; a successful synthesis could stem from factors other than the model's stated logic. Furthermore, the lack of clarity on how these specific cases were selected makes it difficult to determine whether they are representative of the model's typical performance. This introduces a potential for selection bias, which somewhat tempers the strength of the experimental evidence.

---

> ### Author Rebuttal · Authors · 2026-03-31
>
> >**Q1: Reasoning Reliability Evaluation**
>
> We thank the reviewer for raising this important point. We conducted additional evaluations to show that the reasoning process causally contributes to final predictions, rather than merely serving as post-hoc rationalization.
>
> (1) **Reasoning Correctness**. We randomly selected 500 samples and asked three chemistry PhD in organic synthesis to rigorously evaluate the model's reasoning, judging whether the reasoning is correct. The table below shows that the proportion of sound reasoning is higher for correct predictions than for incorrect ones, indicating a strong correlation between answer correctness and reasoning correctness.
>
> |Answer|Reasoning|Coverage (%)|
> |-|-|-|
> |correct|chemically sound|82.85|
> |correct|partially flawed|17.15|
> |wrong|plausible|33.60|
> |wrong|flawed|66.40|
>
> (2) **Decision Consistency**. We examined the correctness of the intermediate decisions (i.e. reaction type, center, reactants) in the model's reasoning on the test set. The results indicate that the correctness of intermediate decisions is well aligned with the final answer. Table 9 further confirms that predictive performance declines when decision space information is removed, indicating that the model relies on this reasoning process.
>
> |Type Decision|Center Decision|Reactants Decision|Final Answer|Coverage (%)|
> |-|-|-|-|-|
> |True|True|True|True|64.06|
> |True|False|True|True|2.84|
> |False|True|True|True|8.79|
> |True|True|False|False|0.38|
> |True|False|False|False|4.57|
> |False|True|False|False|2.01|
> |False|False|False|False|17.35|
>
> These results demonstrate that our method is not merely producing post hoc explanations. Instead, it explicitly guides the LLM to generate structured reasoning before arriving at the final answer.
>
>
> >**Q2: Computational Efficiency & Deployment Feasibility**
>
> We thank the reviewer for highlighting this important concern. For training cost, all experiments were conducted on 8×A800 (40 GB) GPUs. In the single step setting, training on 9k samples took about two days. In the multi step setting, mixed training on 9k single step samples and 20k multi step samples took about six days. We report the inference latency in the table below.
>
> Retro-Expert performs a single round of LLM-based chemical reasoning within the decision space. For single-step setting, the decision space is built from frozen specialized models: T5Chem for reaction type, GraphRetro for reaction center, and Graph2Edits for reactants. For multi-step setting, the LLM is not invoked at every single-step node expansion. We utilize T5Chem to generate the reaction type candidate and DMS-Deep to generate Top-K complete multi-step candidate routes via beam search. These candidates are then jointly provided to the LLM for a single round of reasoning.
>
> As shown in the table, Retro-Expert generates both reasoning and final predictions with modest computational overhead, whereas conventional models produce only the final answers. This makes the proposed synthetic routes more interpretable, and provides practical value by facilitating expert-in-the-loop decision-making.
>
> |Setting|LLM|Specialized Models time / sample (s) ↓|LLM Reasoning time / sample (s) ↓|Total Latency / sample (s) ↓|
> |-|-|-|-|-|
> |single-step|ChemDFM |0.36|2.81|3.17|
> |single-step|ChemLLM |0.36|3.83|4.19|
> |single-step|Ours|0.36|3.26|3.62|
>
>
> |Setting|Method|Total Inference Time / sample (s) ↓|
> |-|-|-|
> |multi-step|DFPN|59.47|
> |multi-step|MCTS|51.73|
> |multi-step|Retro*|51.16|
> |multi-step|Ours|20.41|
>
>
> >**Q3: Wet-lab Validation**
>
> We agree that the wet-lab experiments validate the feasibility of the synthetic routes from the model's reasoning. More precisely, they support the correctness of key decision steps in the reasoning (i.e. reaction type, center, and reactants). For the two cases, we asked chemists to review the model’s reasoning, and they considered its key chemical analyses logically plausible. For example, ''Aryl bromides are suitable electrophiles...so forming the aryl–thiophene C–C bond is plausible.'' A rigorous assessment of overall reasoning correctness is provided in Reviewer GruX-Q1.
>
> For the wet-lab study, we first collected the Top-4 predictions from Retro-Expert on the test set, and then selected experimental cases based on three criteria:
> (1) **Novelty**: cases with unreported synthetic routes;
> (2) **Experimental feasibility**: only cases with accessible starting materials and manageable time/cost were retained;
> (3) **Diversity**: among feasible candidates, we selected examples that reflect different practical application scenarios.
>
> The proposed two cases illustrate two key application scenarios: discovering a new route and designing an alternative route. Both routes were the model's Top-1 predictions.

---

> > ### Author Rebuttal · Reviewer_GruX · 2026-04-07
> >
> > The authors have addressed the primary technical and practical concerns raised in my initial review. I am raising my score to a Weak Accept.

---

> > > ### Author Response · Authors · 2026-04-07
> > >
> > > We greatly appreciate the reviewer's thoughtful and constructive feedback throughout the review process. Your questions and suggestions are highly valuable not only for improving our analysis of the reasoning process, but also helping us present the practical utility of the work more clearly. We will incorporate these improvements in the final revision. Thank you again for your careful reading and valuable guidance.

---

### Decision · Program_Chairs · 2026-04-30

**Decision:**

Accept (regular)

**Comment:**

This paper outlines RetroExpert – a collaboration-based retrosynthesis framework, which combines specialized models with an LLM-based top-level reasoner to aggregate all information and make the final prediction.

On the positive side, reviewers praised strong validation, which includes not only quantitative experiments but also qualitative analyses and wet-lab tests. They considered RetroExpert shows good fusion between the generality of LLMs and high accuracy of domain-specific models, and that it is generally pragmatic and well-executed.

On the negative side, reviewers criticized the multi-step side of the manuscript, which in the initial version of the paper was minimal. To improve this, during rebuttal authors added further experiments, some reviewers considered them already satisfactory, although there also was some disagreement on that front. Moreover, during rebuttal authors have also provided further valuable analyses, notably human validation of the reasoning trace correctness, and results with larger model sizes.

In the end, three reviewers were voting to accept this work, and one was voting to reject (it is worth noting one of the initially negative reviewers mentioned they want to update their score to a "Weak Accept" after rebuttal, yet did not put that into the system).

Having reviewed all information, I believe that, after the improvements during rebuttal, this work is now in a good state, and a decent contribution to ICML. I recommend acceptance.